# Molecular mechanism of BMP signal control by Twisted gastrulation

Tomas Malinauskas [1,9] ✉, Gareth Moore [2,9], Amalie F. Rudolf[1,9],
Holly Eggington[3,4], Hayley L. Belnoue-Davis [3,4], Kamel El Omari [5],
Samuel C. Griffiths [1,7], Rachel E. Woolley[1,8], Ramona Duman[5], Armin Wagner [5],
Simon J. Leedham[3,4], Clair Baldock [6], Hilary L. Ashe [2] ✉ &
Christian Siebold [1] ✉

Twisted gastrulation (TWSG1) is an evolutionarily conserved secreted glyco-protein which controls signaling by Bone Morphogenetic Proteins (BMPs). TWSG1 binds BMPs and their antagonist Chordin to control BMP signaling during embryonic development, kidney regeneration and cancer. We report crystal structures of TWSG1 alone and in complex with a BMP ligand, Growth Differentiation Factor 5. TWSG1 is composed of two distinct, disulfide-rich domains. The TWSG1 N-terminal domain occupies the BMP type 1 receptor binding site on BMPs, whereas the C-terminal domain binds to a Chordin family member. We show that TWSG1 inhibits BMP function in cellular signaling assays and mouse colon organoids. This inhibitory function is abolished in a TWSG1 mutant that cannot bind BMPs. The same mutation in the *Drosophila* TWSG1 ortholog Tsg fails to mediate BMP gradient formation required for dorsal-ventral axis patterning of the early embryo. Our studies reveal the evolutionarily conserved mechanism of BMP signaling inhibition by TWSG1.

Bone morphogenetic proteins (BMPs) are secreted extracellular proteins that form the largest group of the transforming growth factor (TGF) β superfamily. BMPs orchestrate the development and homeostasis of all multicellular organisms. Dimeric BMP ligands initiate signaling by binding to two types of transmembrane receptors: the BMP type 1 (BMPR1) and type 2 (BMPR2) serine/threonine kinases. Bridging both receptors triggers phosphorylation of BMPR1 by constitutively active BMPR2, which in turn activates the SMAD1/5/9 transcription factors[1,2], and ultimately leads to transcriptional control of target genes in the nucleus.

The interaction of BMPs with their receptors is tightly controlled by extracellular antagonists that play key roles during embryogenesis and in maintaining tissue homeostasis. Disturbing the BMP ligand-antagonist balance can lead to developmental diseases and cancer. BMP antagonists form at least four distinct families: Sclerostin, Cer-berus, and DAN (CAN); Noggin; Follistatin; and Chordin[3]. Gremlin-2 (member of the CAN family)[4], Noggin[5], and Crossveinless 2 (member of the Chordin family)[6] directly bind to BMP ligands, compete with their receptors and thus inhibit signaling. Recently, Repulsive Guidance Molecules (RGMs) have emerged as both agonists[7–10] and antagonists[11] of BMP signaling. An anti-RGMA monoclonal antibody, Elezanumab

[1]Division of Structural Biology, Wellcome Centre for Human Genetics, Nuffield Department of Medicine, University of Oxford, Oxford OX3 7BN, UK. [2]Faculty of Biology, Medicine and Health, University of Manchester, Manchester M13 9PT, UK. [3]Intestinal Stem Cell Biology Lab, Wellcome Centre for Human Genetics, University of Oxford, Oxford OX3 7BN, UK. [4]Translational Gastroenterology Unit, John Radcliffe Hospital, University of Oxford, Oxford National Institute for Health Research Biomedical Research Centre, Oxford, UK. [5]Diamond Light Source, Harwell Science and Innovation Campus, Didcot OX11 0DE, UK. [6]Wellcome Centre for Cell-Matrix Research, Faculty of Biology, Medicine and Health, Manchester Academic Health Science Centre, University of Manchester, Manchester M13 9PT, UK. [7]Present address: Evotec (UK) Ltd., 90 Innovation Drive, Milton Park, Abingdon OX14 4RZ, UK. [8]Present address: Etcembly Ltd., Atlas Building, Harwell Campus OX11 0QX, UK. [9]These authors contributed equally: Tomas Malinauskas, Gareth Moore, Amalie F. Rudolf. ✉e-mail: tomas@strubi.ox.ac.uk; hilary.ashe@manchester.ac.uk; christian@strubi.ox.ac.uk

(Abbott), is being evaluated as a treatment for multiple sclerosis, spinal cord injury, and ischemic stroke.

Twisted gastrulation (Tsg in *Drosophila*, TWSG1-a/-b in *Xenopus*, TWSG1 in humans) is a secreted glycoprotein that is required, together with Chordin family members (Short Gastrulation (Sog) in *Drosophila*), for BMP gradient formation during dorsal-ventral patterning of invertebrate and vertebrate embryos[12–16]. In this context, TWSG1 can function as both an agonist and antagonist of BMP signaling. TWSG1 can inhibit BMP signaling by either sequestering BMPs directly from their receptors[14,17] or promoting the formation of an inhibitory Chordin–BMP–TWSG1 ternary complex[12,14,15]. However, with no structural data available for TWSG1 or any of these interactions, its precise mechanism of action remains obscure.

Here, we present the high-resolution structure of full-length human TWSG1 comprising distinct N- and C-terminal domains (NTD and CTD) that show no homology to other protein structures. We show that the TWSG1 NTD binds to several BMP ligands, whereas the CTD interacts with Chordin. We also determined the structure of TWSG1 NTD in complex with the BMP ligand GDF5 revealing that TWSG1 binds to the BMPR1-binding site on GDF5. Structure-guided TWSG1–BMP-binding experiments, combined with cellular BMP signaling assays and experiments using mouse colon organoids, confirm the observed TWSG1–GDF5 interaction determinants. Finally, we show that introducing the *tsg* point mutants, which impair interactions with BMP ligands, into *Drosophila* results in a loss of embryonic BMP gradient formation. Taken together, these studies illuminate the evolutionarily conserved mechanism of BMP signaling inhibition by TWSG1 and enable future functional and translational studies to decipher its pro- and anti-BMP signaling roles.

## Results

To enable biophysical and cellular studies of human TWSG1, we produced full-length glycosylated TWSG1 (TWSG1, Cys26–Phe223, Fig. 1A) by transient transfection of human embryonic kidney (HEK) 293 T cells. We crystallized and determined the TWSG1 structure at 2.6 Å resolution using a platinum single-wavelength anomalous diffraction experiment (Supplementary Table 1). TWSG1 crystallized as a dimer in the asymmetric unit, an arrangement that we observed in two crystal forms (Supplementary Table 1). This dimeric arrangement agrees with size-exclusion chromatography (SEC) coupled with multi-angle light scattering (SEC-MALS) carried out in solution, which suggests a monomer-dimer equilibrium (Fig. 1B).

The structure of TWSG1 reveals a modular architecture, comprising an α-helical N-terminal domain (NTD, Cys26–Arg80) connected by an extended 18 Å-long linker to a mixed-α/β C-terminal domain (CTD, Pro87–Phe223) (Fig. 1C). Search of structural homologs among all previously determined structures in the Protein Data Bank (PDB) did not yield any hits highlighting TWSG1's unique structural folds and their arrangement. The NTD of TWSG1 folds into a compact bundle of three α-helices locked by 7 disulfides (Fig. 1D). The CTD of TWSG1 is stabilized by 5 disulfide bonds (VIII–XII) and contains a sheet comprised of 5 antiparallel β-strands which constitute its core (Fig. 1E). Detailed analysis of TWSG1 architecture is provided in the Supplementary Discussion. Taken together, the crystal structure of human TWSG1 reveals two previously unseen, evolutionarily conserved disulfide-rich domains and points towards several surface-exposed residues that could modulate the BMP signaling.

### The N- and C-terminal domains of TWSG1 interact with the BMPs and CHRDL2

Based on the observed TWSG1 two-domain architecture, we designed protein constructs encompassing either the NTD or CTD, and dissected their BMP-binding properties. We performed surface plasmon resonance (SPR)-based equilibrium binding experiments using purified proteins. First, we used the three human BMP ligands, BMP7 (TWSG1 ligand in kidney[18]), GDF5, and BMP2 (the latter two are ubiquitously expressed in adults). BMP ligands were immobilized on the SPR chip surface, and TWSG1 NTD or CTD (or bovine serum albumin, control) were used as analytes. The TWSG1 NTD bound to BMP7 and GDF5 with equilibrium dissociation constants ($K_d$s) of 0.094 and 0.38 μM, respectively, and slightly weaker to BMP2 (with a $K_d$ of 1.015 μM) (Fig. 1F–H, and Supplementary Fig. 2). In contrast, the TWSG1 CTD bound to all three BMPs notably (~100-fold) weaker (Fig. 1F–H). Thus, our SPR data point to the NTD as the major mediator of TWSG1–BMP interactions. This agrees with a previous study showing that a construct comprising the N-terminal region of TWSG1 from *Xenopus* and *Drosophila* is sufficient to bind to BMP4 in immunoprecipitation assays[13]. Next, we carried out the same experiment with the Chordin superfamily member CHRDL2. Previous studies have shown that full-length TWSG1 directly binds to CHRDL2[19], and that a mutation in the *Xenopus* TWSG1 CTD abolishes interactions with Chordin[17,20]. Our data show that, in contrast to BMP ligands, CHRDL2 bound exclusively to the TWSG1 CTD ($K_d$ 0.46 μM, Fig. 1I and Supplementary Fig. 2). We validated our SPR data using SEC-MALS, where we observed stable complexes of CHRDL2 with either the TWSG1 CTD or full-length TWSG1. Interestingly, CHRDL2 showed some propensity for dimerization, similar to TWSG1 (Supplementary Fig. 1I, J). Taken together, our biophysical analyses reveal that TWSG1 acts as a platform for both BMP ligands and Chordin family members.

### Structure of the TWSG1 N-terminal domain in complex with BMP ligand GDF5

To understand how TWSG1 interacts with BMP ligands, we determined the crystal structure of the TWSG1 NTD in complex with GDF5 at 1.96 Å resolution (Fig. 2A and Supplementary Table 1). The disulfide-linked GDF5 dimer binds to two TWSG1 NTD molecules that are related by a non-crystallographic pseudo two-fold axis (RMSD of 0.419 Å for 138 equivalent Cα atoms of the 1:1 TWSG1 NTD:GDF5 complexes). The two TWSG1 NTD molecules in the complex adopt a similar conformation compared to the apo TWSG1 NTD (0.33 Å for 38 equivalent Cα atoms), with the major variability observed in the loop connecting helices α1 and α2. The TWSG1 NTD binds to the finger region of GDF5 and interacts with both protomers of the GDF5 dimer. The interface between TWSG1 (dark orange in Fig. 2A) and GDF5 buries a total surface area of 1975 Å².

TWSG1–GDF5 interactions are mediated by two salt bridges (TWSG1 Lys28-GDF5 Glu434 and TWSG1 Glu42-GDF5 Lys488), 4 hydrogen bonds, and 72 hydrophobic contacts involving 20 residues from TWSG1 and 18 residues from GDF5. The TWSG1–GDF5 interface centers around helix α1 of TWSG1 sequestered between the two finger-like motifs and helix α3 of GDF5 (Fig. 2A–C). The side chain of TWSG1 Ile40, at the core of this interface, inserts into the hydrophobic pocket of GDF5 formed by Trp414, Trp417, Val448, Ile449, Phe478, and Tyr490 (Fig. 2B), and is shielded by a salt bridge between TWSG1 Glu42 and GDF5 Lys488. The side chain of GDF5 Phe435 further anchors helix α1 of TWSG1 by hydrophobic interactions with TWSG1 Ala32, Val35, and Lys28. The salt bridge between TWSG1 Lys28 and GDF5 Glu434 shields this hydrophobic interface.

Unexpectedly, the structure of the TWSG1–GDF5 complex reveals a previously unknown calcium-binding site at the edge of TWSG1–GDF5 interface (Fig. 2C). The calcium ion is coordinated by 8 oxygen atoms with an average distance of 2.55 Å: carboxyl group of GDF5 Asp416, main-chain carbonyl oxygen of GDF5 Gly413 and 5 water molecules. 2 out of 5 water molecules bridge the calcium ion to the carboxyl group of TWSG1 Asp34. The calcium-binding site is stabilized by a hydrogen bond between TWSG1 Ser33 and GDF5 Trp414, and other interactions (salt bridge between TWSG1 Asp34 and Lys37; T-shaped π-π stacking of Trp414 and Trp417 of GDF5). We confirmed the identity of the calcium ions by anomalous difference Fourier analysis (Supplementary Fig. 3). GDF5 Gly413 and Asp414 are conserved in 18 out of 20, and 12 out of 20

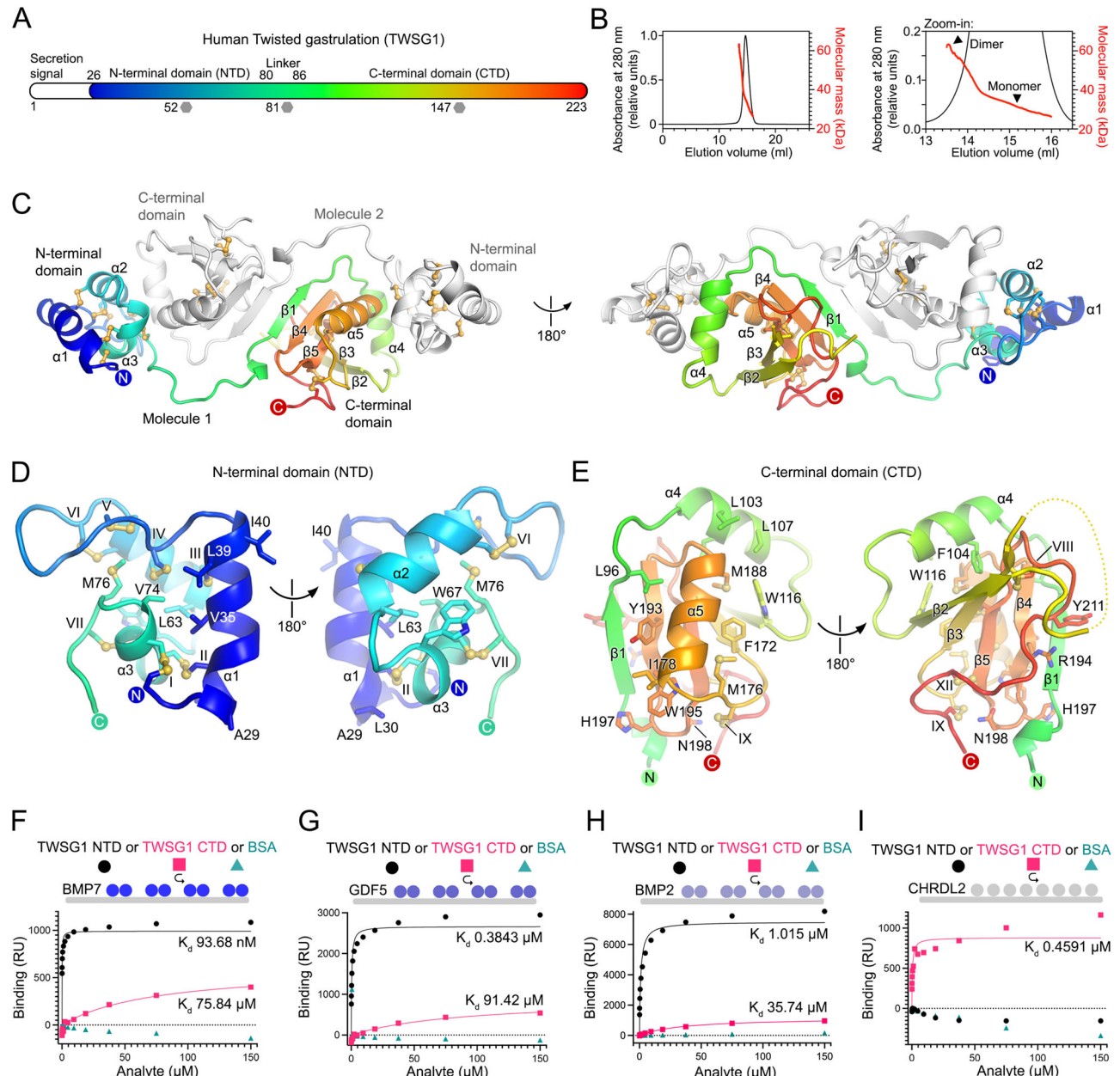

**Fig. 1 | Crystal structure of human twisted gastrulation (TWSG1). A** Domain organization of human TWSG1. Predicted glycosylated asparagine residues (52, 81, and 147) are marked with gray hexagons. **B** SEC-MALS analysis of TWSG1. The experimentally determined molecular mass of TWSG1 varies from 26,332 to 61,937 Da. Theoretical molecular mass of the TWSG1 monomer is 23,541 Da protein plus 5648 Da Asn-linked glycans (three $Man_9GlcNAc_2$ moieties, 1.88264 kDa each). Traces of absorbance at 280 nm and calculated molecular mass are colored in black and red, respectively. **C** Crystal structure of the human TWSG1 dimer with one protomer colored as rainbow (N-terminus, blue; C-terminus, red) and one in gray. The two views differ by a 180° rotation around a vertical axis. **D, E** Architecture of the TWSG1 N- and C-terminal domains (NTD and CTD). Disulfide bonds are numbered in Roman numerals and shown as yellow spheres and sticks. **F–I** SPR-based equilibrium binding experiments. Different concentrations of TWSG1 NTD (black circles), CTD (pink squares), and BSA (aquamarine triangles) were injected over surfaces coated with BMP7 (**F**), GDF5 (**G**), BMP2 (**H**), or CHRDL2 (**I**). Equilibrium binding dissociation constants ($K_d$s) are indicated. RU response units.

human BMP family members, respectively[11]. As the local concentration of calcium in the extracellular space can vary significantly (~1.0–2.0 mM)[21,22], calcium might modulate interactions between BMP family members, their receptors, agonists, and antagonists. For this reason, we carried out our SPR-based binding studies in the presence of 2 mM $CaCl_2$ to reflect extracellular calcium levels.

### TWSG1 occupies the BMP type 1 Receptor-binding site
Structures of BMP ligands in complex with their receptor ectodomains revealed a common architecture, in which BMPR1 binds to a dimeric BMP ligand independently of BMPR2, assembling into a 2:2:2 BMP:BMPR1:BMPR2 hexameric complex[23–25]. In our TWSG1–GDF5 complex, the TWSG1 NTD occupies the BMPR1-binding site on GDF5 (Fig. 2D). This so-called "wrist" epitope of the BMP ligand also binds to the RGM co-receptors (Fig. 2E)[10,11], and the BMP9 pro-domain (Fig. 2F)[26]. Strikingly, TWSG1, RGMs, BMP-prodomains, and the BMPR1 ectodomains evolved the same structural mechanism to mediate interactions with BMP ligands. They insert an α-helix into the hydrophobic groove formed by two fingers of one BMP protomer and the α-helix of the second protomer (Fig. 2G–I). The inserted α-helix is anchored by a hydrophobic "finger" residue (Ile40 in TWSG1) inserted into the hydrophobic cavity of the BMP ligand (Fig. 2B, G–I).

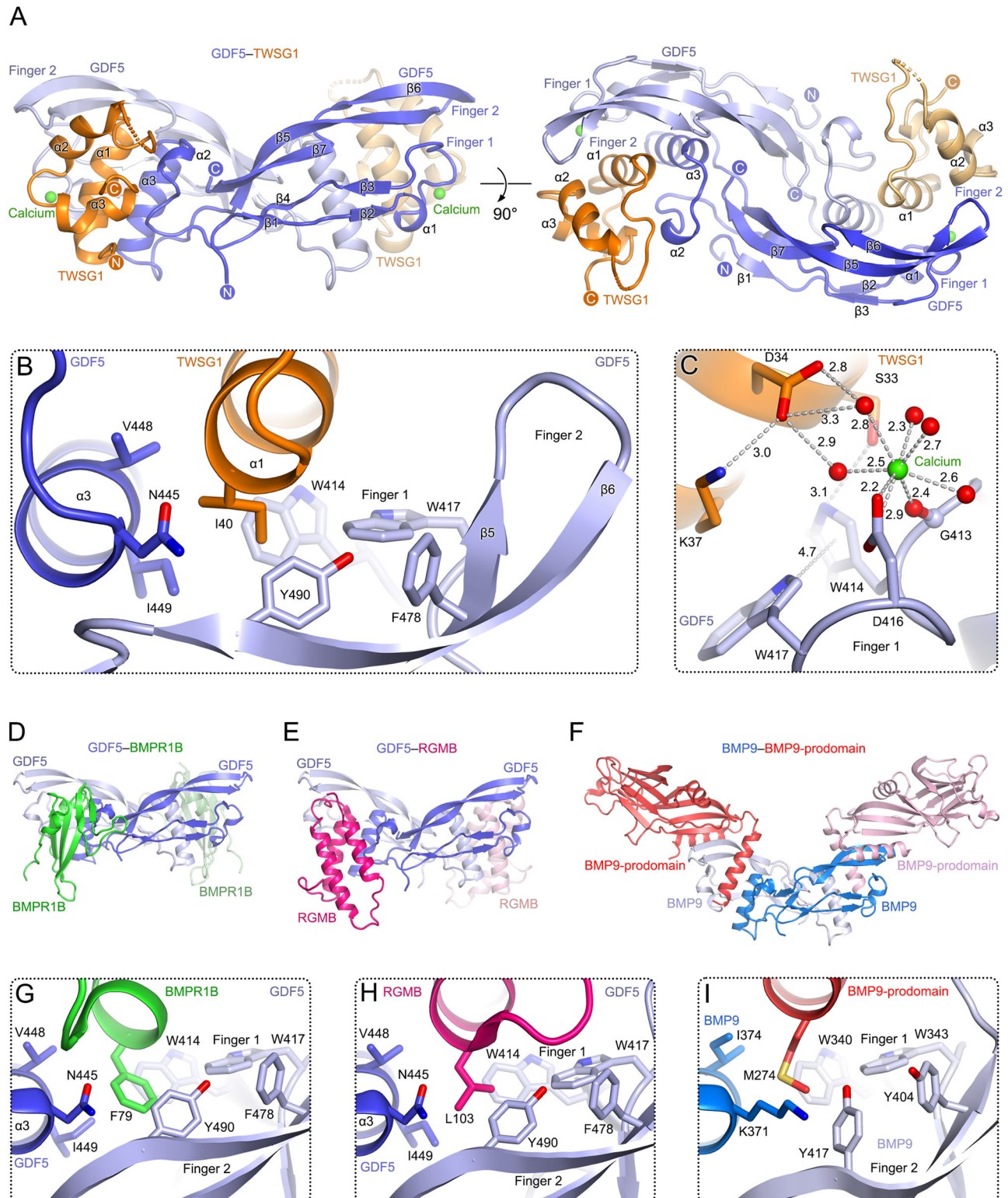

**Fig. 2 | Crystal structure of human TWSG1 in complex with Growth differentiation factor 5 (GDF5). A** Crystal structure of the TWSG1–GDF5 complex. The two views of the dimer differ by a 90° rotation around a horizontal axis. The disulfide-linked GDF5 dimer is colored in dark and light blue. The two TWSG1 NTDs are colored in orange and wheat. Calcium ions are shown as green spheres. **B** The side chain of TWSG1 Ile40 inserts into the hydrophobic pocket formed by two GDF5 protomers. Side chains of key residues are shown as sticks with oxygen and nitrogen atoms colored in red and blue, respectively. **C** Calcium-binding site forming interface contacts between TWSG1 and GDF5. Water molecules are shown as red spheres. Distances (Å) between selected atoms and GDF5 Trp414-Trp417

rings (T-shaped π-π stacking) are indicated with gray dashed lines. **D**–**F** Structures of GDF5 in complex with the BMPR1B ectodomain (**D**; PDB ID 3EVS)[25], the co-receptor RGMB (**E**; PDB ID 6Z3J)[11], and BMP9 in complex with its pro-domain (**F**; PDB ID 4YCG)[26]. GDF5 and BMP9 are shown in the same orientation as in **A**. **G**–**I** Close-up views of the interfaces for the complexes shown in **D**–**F**. In all cases, interactions are mediated by a hydrophobic side chain exposed on the α-helix of the GDF5/BMP9-binding partner and a hydrophobic pocket formed by the two GDF5/BMP9 protomers. This mode of interaction is also observed in the TWSG1–GDF5 complex (shown in **B**).

To further investigate the TWSG1–GDF5 interface in our structure, we performed structure-guided mutagenesis of the key interface residue TWSG1 Ile40 (Fig. 2B) and tested binding to three BMP ligands. TWSG1 Ile40Ala–BMP7 interactions were non-detectable when BMP7 was immobilized at lower level (137 RU) on the SPR chip or were notably weakened when BMP7 was immobilized at higher level (7372 RU, $K_d$ 454.4 µM) (Supplementary Fig. 4A, B). Similarly, interactions between TWSG1 Ile40Ala and either GDF5 or BMP2 were largely abolished (Supplementary Fig. 4C–F). These data suggest that TWSG1 engages diverse BMP ligands in a conserved mechanism with Ile40 at the center of interactions.

Next, we investigated whether calcium contributes to TWSG1–BMP interactions in solution as suggested by our crystal structure (Fig. 2C). We mutated the calcium-binding Asp34 of TWSG1 to alanine and tested binding to BMP7, GDF5, and BMP2 (Supplementary Fig. 4G–L). The TWSG1 mutation Asp34Ala did not abolish BMP interactions, resulting in $K_d$s comparable to wild-type TWSG1 (Fig. 1F–H and Supplementary Fig. 4G–L), suggesting that calcium plays a minor role in TWSG1–BMP interactions. Since residues equivalent to calcium-binding Gly413 and Asp416 of GDF5 are highly conserved across BMP family members[11] (Fig. 2C), we investigated whether calcium could affect interactions of BMP ligands with BMPR1B, the type 1 receptor that engages GDF5 during bone and joint formation[25]. EDTA weakened interactions of BMPR1B with all tested BMP ligands (GDF5, BMP2 and BMP7) leading to a ~2–5-fold increase of $K_d$ values (Supplementary Fig. 5). Taken together, these data suggest that calcium could modulate interactions between BMPs and their binding partners. However, further studies are required to unravel the complex interactions of BMP ligands with their receptors and modulators.

## TWSG1 inhibits BMP signaling in cellular assays

In order to relate our structural and biophysical binding experiments to cellular signaling, we conducted a BMP-responsive luciferase reporter assay in C2C12 myoblast cells, which is responsive to many BMP family members, including GDF5, BMP2, and BMP7[10,11,27]. We first focused on the TWSG1–GDF5 interaction because both proteins share overlapping expression patterns (based on the Human Protein Atlas[28]). GDF5 readily activated signaling in a concentration-dependent manner with a half-maximum concentration ($EC_{50}$) of 26.63 nM (Fig. 3A and Supplementary Fig. 6A). Addition of purified TWSG1 protein (at 1 µM concentration) effectively inhibited signaling across a broad concentration range of GDF5 (up to 1.97 µM). Specifically, TWSG1 inhibited GDF5 signaling (at 40 nM concentration) in a concentration-dependent manner with a half-maximum inhibitory concentration ($IC_{50}$) of 67.11 nM (Fig. 3B and Supplementary Fig. 6B). Finally, we compared TWSG1 to Gremlin-1, a conserved, secreted inhibitor of BMP signaling[29]. Indeed, TWSG1 was as effective in inhibiting GDF5 signaling as Gremlin-1, thus suggesting that TWSG1 acts as a potent inhibitor of BMP signaling in C2C12 myoblasts (Fig. 3C).

Next, we tested the effect of TWSG1 on BMP2 and BMP7 signaling. Similar to GDF5, BMP2 and BMP7 signaling was readily activated in C2C12 myoblasts (10–40 nM GDF5 or BMP2; 5–20 nM BMP7) (Fig. 3D–F). However, only GDF5 and BMP7 signaling was efficiently inhibited by wild-type TWSG1 (1 µM), and no inhibition was observed for BMP2. This agrees with previous observations showing that TWSG1 effectively inhibited signaling by BMP7 but not by BMP4, which shares 95% sequence identity with BMP2[30] and confirms that TWSG1 inhibits signaling by specific BMP ligands. To further validate the TWSG1–BMP interface observed in our structure, we investigated the role of Ile40, a key interface residue crucial for BMP binding (Fig. 2B), in our signaling assay. Mutation of Ile40 to either alanine or glutamate abolishes the inhibitory function of TWSG1 in GDF5 and BMP7 signaling (Fig. 3D, E, and Supplementary Fig. 6). Finally, the TWSG1 NTD (but not NTD Ile40Ala or CTD) inhibits GDF5 signaling as efficiently as the full-length TWSG1 (Supplementary Fig. 7).

In addition to monitoring BMP-responsive transcription, we visualized pSmad levels in C2C12 cells treated with BMP2, BMP7, or GDF5. For each BMP ligand, the effect of adding wild-type or mutant TWSG1 was determined. Consistent with the luciferase assay results (Fig. 3C–F), BMP2 treatment led to an increase in pSmad1/5/9 levels, and the addition of TWSG1 proteins had no significant effect (Supplementary Fig. 8A–B). However, pSmad1/5/9 accumulation in response to either BMP7 or GDF5 was inhibited by wild-type TWSG1, whereas no inhibition was observed upon TWSG1 Ile40Glu addition (Supplementary Fig. 8C–D). In these experiments, GDF5 addition resulted in lower pSmad1/5/9 levels than BMP2 and BMP7 (Supplementary Fig. 8D–E), as described previously[31].

Finally, we exploited the ability of BMP signaling to promote differentiation of C2C12 myoblasts into osteoblasts[32] to further explore TWSG1's inhibitory function. We used BMP7 as the ligand in this experiment because, based on the pSmad quantitation and luciferase assay data, BMP7 strongly activated the pathway and was inhibited by TWSG1 (Fig. 3E, Supplementary Fig. 8C). In the absence of BMP7, C2C12 cells differentiate into myotubes that are positive for Myosin heavy chain IV as visualized by immunofluorescence (Supplementary Fig. 9)[32]. However, treatment with BMP7 inhibits myotube formation and instead promotes differentiation into Alkaline Phosphatase-positive osteoblasts. Wild-type TWSG1 inhibits the ability of BMP7 to promote differentiation into osteoblasts, resulting in the formation of myotubes. In contrast, osteoblast differentiation is observed upon the addition of BMP7 and TWSG1 Ile40Glu, consistent with the mutation abolishing TWSG1's ability to inhibit BMP7 signaling (Supplementary Fig. 9). Taken together, these data confirm that the TWSG1–BMP interactions observed in the structure play a key role in the cellular context and highlight the critical role of TWSG1 Ile40 for the assembly of the TWSG1–BMP complex.

## TWSG1 promotes the growth of intestinal organoids by inhibiting BMP signaling

Multiple antagonists of BMP signaling are expressed in intestinal tissues and regulate their homeostasis[29], for example human TWSG1 that is expressed in human intestinal organs (Human Protein Atlas[28]). Gastric intestinal metaplasia is inhibited by upregulation of TWSG1 expression[33]. We assessed the potential of TWSG1 to promote growth of organoids derived from mouse intestinal crypts, which have been successfully used to study BMP signaling antagonists such as Gremlin-1, and as a model system for colon cancer[29,34] (Fig. 3G).

Development of organoids from isolated intestinal crypts and continuous maintenance in culture requires at least three growth factors: Epidermal Growth Factor (EGF), an antagonist of BMP signaling (e.g., Noggin or Gremlin-1), and R-spondin 1 (an agonist of Wnt signaling) (ENR media). Loss of BMP antagonism becomes phenotypically visible as loss of organoid survival throughout successive passages in culture. In murine intestinal organoids, the predominant BMP ligands are murine Bmp7 and Bmp2, whereas Bmp4 is expressed at a lower level (Fig. 3H). Thus, this analysis suggests that antagonism of Bmp7 and Bmp2 is required to maintain organoids in proliferative and non-differentiated states.

In the presence of R-spondin-1 and EGF but without a BMP antagonist, organoids showed a depleted number over subsequent passages with spheroid-like structures lacking proper budding and outgrowth (Fig. 3I, ER media, negative control; Supplementary Fig. 10). In the presence of complete ENR media, organoids demonstrated strong survival, continued growth, and budding (Fig. 3J, ENR media, positive control). To test whether TWSG1 affects BMP signaling in intestinal organoids, we replaced Noggin with our purified TWSG1 (Fig. 3K, ER plus TWSG1). Organoid survival was rescued, and widespread budding was restored when organoids were grown in the presence of EGF, R-spondin-1, and wild-type TWSG1 over three consecutive passages, indicating that TWSG1 could function as an

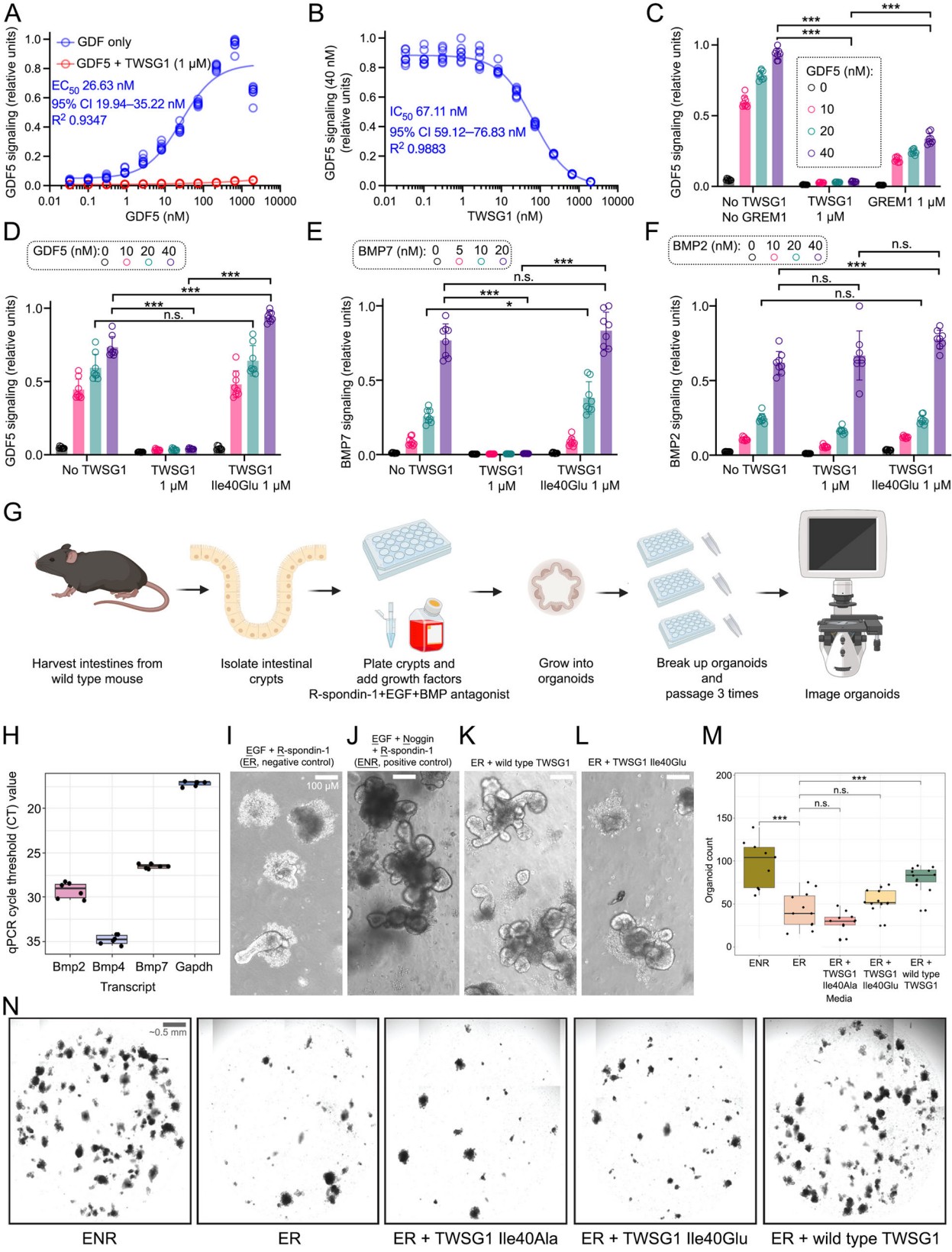

antagonist of BMP signaling in mouse intestinal cells. Mutation of TWSG1 Ile40 to glutamate or alanine impaired TWSG1 ability to rescue organoid survival (Fig. 3L, M and Supplementary Fig. 10D, E). Our cellular signaling assays and organoid experiments show that wild-type (but not mutant) TWSG1 acts as an inhibitor of BMP signaling.

## The evolutionarily conserved TWSG1–BMP interactions in *Drosophila*

Since TWSG1 plays a critical role in BMP gradient formation[35] (Fig. 4A), we extended our structure-function analysis to dorsal-ventral patterning in the early *Drosophila* embryo. First, we used a tissue culture-based assay to test the ability of wild-type Tsg and Tsg[I40E] to interact

**Fig. 3 | TWSG1 inhibits BMP signaling in cellular assays and mouse intestinal organoids. A** GDF5 activates SMAD-dependent BMP signaling in a concentration-dependent manner in C2C12 myoblasts with a half-maximal effective concentration ($EC_{50}$) of 26.63 nM (blue circles). TWSG1 (1 µM) inhibited GDF5 signaling (red circles). **B** TWSG1 inhibits GDF5 (40 nM) signaling in a concentration-dependent manner in C2C12 myoblasts with a half-maximal inhibitory concentration ($IC_{50}$) of 67.11 nM. **A–C**, Activation of a luciferase reporter assay as a readout of GDF5 signaling was measured eight times at each GDF5 concentration ($n = 8$, indicated by open circles). CI, Confidence Interval. XY signaling data were fitted to a non-linear, sigmoidal, four-parameter logistic model. **C** Comparison of TWSG1 and Gremlin-1 (GREM1), a BMP signaling inhibitor. Both TWSG1 (1 µM) and GREM1 (1 µM) inhibited GDF5 signaling (0, 10, 20, and 40 nM), but TWSG1 seemed to be a stronger inhibitor than GREM1. **D, E** Wild-type TWSG1 (but not TWSG1 Ile40Glu) inhibits BMP5 (**D**) and BMP7 (**E**) signaling. **F** Neither wild-type TWSG1 nor TWSG1 Ile40Glu inhibits BMP2 signaling. **C–F** Each column represents the average signaling, measured eight times ($n = 8$, indicated by open circles). Standard deviations are indicated by vertical T-shaped bars on each column. $P$ values were calculated using Student's two-sample, two-tailed $t$ test, assuming unequal variance: n.s., not significant, $P > 0.05$; *$P \leq 0.05$; **$P \leq 0.01$; ***$P \leq 0.001$. Exact $P$ values are presented in the Source Data file. **G** Overview of the preparation of mouse intestinal organoids, their growth in the presence of BMP signaling antagonists, and image analysis. **G** was created with BioRender.com released under a Creative Commons Attribution-NonCommercial-NoDerivs 4.0 International license https://creativecommons.org/licenses/by-nc-nd/4.0/deed.en **H** qPCR data shows that Bmp7 and Bmp2 are predominantly expressed by intestinal organoids, while Bmp4 is expressed at a lower level ($n = 6$, number of wells). CT value lower than 30 indicates notable protein expression. Data are presented in the form of a box plot, where: outer bounds (whiskers) indicate minima and maxima with 1.5 times the interquartile range, inner bounds indicate interquartile range (25th–75th percentile), and central line indicates 50th percentile (median) of data. Each qPCR reaction was carried out in duplicate and mean value is plotted. **I** Intestinal organoids grow poorly in the presence of Epidermal Growth Factor (EGF) plus R-Spondin-1 (agonist of the Wnt signaling pathway) (negative control, ER media), but (**J**) demonstrate enhanced growth, survival, and budding when EGF and R-Spondin-1 are supplemented with a BMP signaling antagonist Noggin (positive control, ENR medium). **K** Noggin could be replaced with TWSG1, suggesting that TWSG1 can function as a BMP antagonist in intestinal organoid cultures. **L** TWSG1 Ile40Glu does not bind to BMP ligands, does not act as a BMP antagonist, and does not support the growth of organoids. **M** Quantification of organoid number post third passage ($n = 9, 11, 12, 12, 12; n = $ number of wells per condition). $P$ values were calculated using Student's $t$ test, two-sample, two-tailed assuming unequal variance. ENR versus ER, ***$P = 2.18 \times 10^{-4}$; ER versus ER plus TWSG1, ***$P = 1.39 \times 10^{-4}$; ER versus ER plus TWSG1 Ile40Glu, n.s., $P = 0.15$; ER versus ER plus TWSG1 Ile40Ala, n.s., $P = 0.067$. n.s., not significant, $P > 0.05$. Data are presented in the form of a box plot, where: outer bounds (whiskers) indicate minima and maxima with 1.5 time the interquartile range, inner bounds indicate interquartile range (25th–75th percentile), and central line indicates 50th percentile (median) of data. **N** Macroscopic imaging of Matrigel domes indicated gross rescue of compromised organoid survival as a result of Bmp antagonist withdrawal across consecutive passages with TWSG1 supplementation, which was not recapitulated by mutant variants. Counts of imaged organoids are presented in **M**.

with the most potent BMP signaling molecule in the early *Drosophila* embryo, the Decapentaplegic (Dpp)–Screw (Scw) heterodimer[36]. Conditioned media was collected from *Drosophila* S2 cells co-transfected with Dpp:HA and Scw:Flag expression plasmids, which promotes heterodimer formation[36,37]. Sog:Myc, Tsg:His and Tsg$^{I40E}$:His plasmids were also each transfected into cells, and various combinations of the Sog and Tsg conditioned media were mixed with the Dpp:HA–Scw:Flag media. Dpp:HA–Scw:Flag heterodimers were then immobilized on anti-Flag beads and the amount of bound Sog or Tsg protein was visualized by Western blotting. Dpp:HA is detected bound to the anti-Flag beads (Fig. 4B), consistent with the formation of Dpp:HA–Scw:Flag heterodimers. Wild-type Tsg bound weakly to Dpp:HA–Scw:Flag heterodimers in the absence of Sog and more strongly when Sog was present. In contrast, Tsg$^{I40E}$ did not bind to Dpp:HA–Scw:Flag, even in the presence of Sog, despite equivalent expression levels of wild-type and mutant Tsg proteins in the inputs (Fig. 4B). In addition, Sog interaction with Dpp:HA–Scw:Flag is reduced in the presence of Tsg$^{I40E}$, compared to the addition of either Sog alone or both Sog and wild-type Tsg (Fig. 4B), raising the possibility that Tsg$^{I40E}$ can sequester Sog and prevent its interaction with Dpp–Scw (see Discussion).

As Tsg$^{I40E}$ is unable to bind to Dpp–Scw, we tested the effect of introducing the Ile40Glu and Ile40Ala mutations in vivo. First, we used CRISPR genome editing to replace the endogenous *tsg* sequences on the X chromosome with an attP landing site[38,39] (Fig. 4C). As expected for a *tsg* null mutation[40], this *tsg*$^{attP}$ genome edit confers male lethality. To further characterize the *tsg*$^{attP}$ mutant embryos, we used single-molecule fluorescent in situ hybridization (smFISH) to visualize the expression of BMP target genes. We focused on the *Race* and *u-shaped* (*ush*) target genes, which respond to peak and intermediate levels of BMP signaling, respectively. As such, *Race* expression is restricted to a narrow stripe in the presumptive amnioserosa of wild-type embryos, whereas *ush* has a broader expression pattern[41] (Fig. 4D). In contrast, embryos carrying the newly generated *tsg*$^{attP}$ allele show a loss of *Race* expression in the presumptive amnioserosa and expanded *ush* expression (Fig. 4D). These defects are characteristic of *tsg* null embryos, which lack a BMP gradient and instead have a uniform low level of signaling in the dorsal ectoderm. This signaling level is sufficient to activate *ush*, which therefore has a broader expression pattern

than in wild-type embryos, but is too low to activate *Race* expression in the presumptive amnioserosa[14,37,42] (Fig. 4A). As a direct comparison, we show that the *Race* and *ush* expression pattern defects in *tsg*$^{attP}$ embryos are the same as those carrying a previously described *tsg*$^2$ null allele[40] (Fig. 4D). Taken together, the defects in BMP target gene expression, along with the male lethality and sequencing data across the *tsg* locus, confirm the deletion of *tsg* sequences in the *tsg*$^{attP}$ flies.

We next made use of the attP landing site in the *tsg* locus to reintegrate wild-type and mutant versions of the *tsg* gene (along with the additional sequences removed by the CRISPR edit) back into the endogenous locus by ΦC31-mediated transgenesis (Fig. 4C). Reintegration of wildtype *tsg* coding sequences with a C-terminal ALFA-tag results in viable male flies. smFISH analysis and quantitation of the width of each expression pattern from the dorsal midline in the center of the embryo reveals that the resulting *tsg:ALFA* embryos have wild-type *Race* and *ush* expression patterns (Fig. 4E–G). Introduction of a mutant version of the *tsg* cDNA carrying either an Ile40Ala or Ile40Glu mutation does not rescue the male lethality. In addition, the *Race* and *ush* expression patterns are disrupted in embryos carrying the *tsg*$^{I40A}$ or *tsg*$^{I40E}$ insertions (Fig. 4E), indicating that these point mutations abrogate Tsg's function in BMP gradient formation. However, we note that the residual anterior *Race* staining in the *tsg*$^{I40A}$ or *tsg*$^{I40E}$ mutant embryos is more intense than in the *tsg* null mutants (Fig. 4D cf. Fig. 4E, see Discussion). Quantitation of the width of the *Race* and *ush* expression domains across the dorsal midline in the center of embryos for all the genotypes analyzed is shown in Fig. 4F–G.

We also visualized the distribution of pMad, the phosphorylated Smad transcription factor downstream of BMP signaling[35], in embryos carrying the *tsg*$^{I40A}$ or *tsg*$^{I40E}$ mutations. Wild-type embryos have a pMad stripe at the dorsal midline[43,44] that is lost in the *tsg*$^{attP}$ and *tsg*$^2$ embryos (Supplementary Fig. 11). The pMad stripe is restored in embryos carrying the wild-type *tsg:ALFA*, but not the *tsg*$^{I40A}$ or *tsg*$^{I40E}$ insertions (Supplementary Fig. 11), providing further evidence that the Tsg Ile40Ala and Tsg Ile40Glu proteins cannot support BMP gradient formation. Together these data show that residue Ile40, which is necessary for BMP interaction in vitro and in signaling and organoid assays, is also critical for Tsg regulation of Dpp–Scw signaling in vivo during embryonic dorsal-ventral patterning.

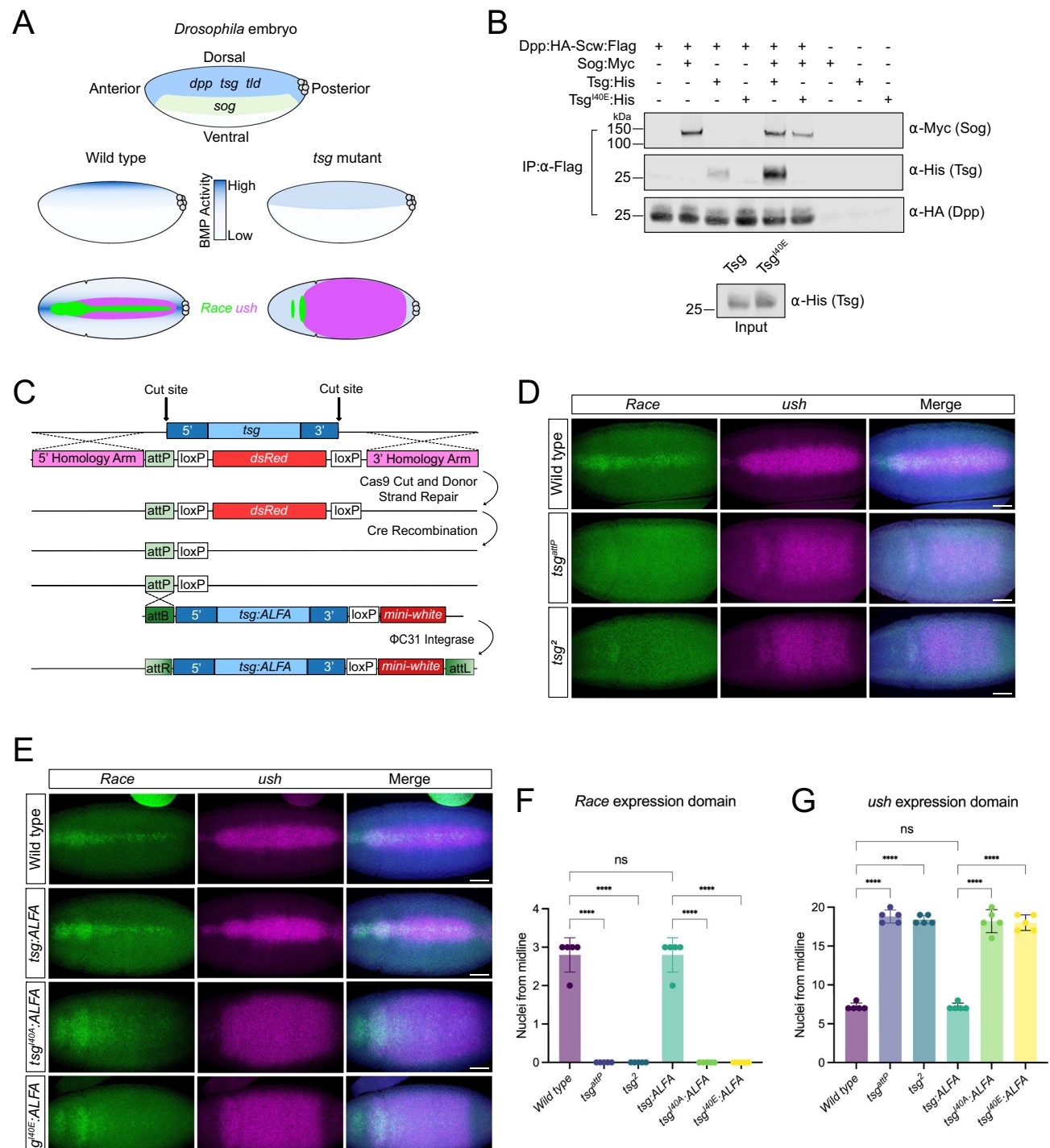

## Discussion

Although mutations in *Drosophila* Tsg were first shown to interfere with embryonic development in 1984[45], the structure of Tsg or any structural homolog has remained unknown. Our structural analysis here revealed a modular architecture of TWSG1 comprising two disulfide-rich domains containing previously unseen folds (Fig. 1). Remarkably, a relatively small NTD of ~50 amino acid residues contains 7 disulfide bonds and thus serves as a stable, rigid scaffold suitable to mediate TWSG1 interactions with its binding partners. The TWSG1 NTD is connected to the C-terminal domain (CTD) via a linker of variable sequence but of relatively constant length (~10 residues) across species and different proteins (e.g., *Drosophila* Crossveinless or proteins in

human parasites, Supplementary Fig. 1). This suggests that both domains and their relative position to each other might be important for TWSG1 interactions with its two key partners, BMP ligands and Chordin family members, and the potential formation of a ternary TWSG1–BMP–Chordin complex.

The modular architecture of TWSG1 enabled us to design constructs of individual domains (NTD and CTD) and dissect both their BMP- and Chordin-binding properties. We identified the TWSG1 NTD as the high-affinity BMP-binding domain, enabling us to determine its structure in complex with a dimeric BMP ligand (Figs. 2 and 5). This structure revealed that TWSG1 NTD binds to the same site on BMP as BMPR1, suggesting that TWSG1 inhibits BMP signaling by competing

**Fig. 4 | The mutation of Tsg Ile40 leads to a loss of BMP gradient formation in the *Drosophila* embryo. A** Cartoon of a lateral *Drosophila* embryo showing that *dpp*, *tsg* and *tld* genes are expressed in the dorsal ectoderm, whereas *sog* is expressed in the neuroectoderm. *scw* is ubiquitously expressed in the embryo. These extracellular proteins lead to a steep BMP activity gradient in the dorsal ectoderm in a wild-type embryo that activates the *Race* and *ush* target genes (middle cartoon: lateral view; bottom cartoon: dorsal view). In *tsg* mutants, BMP gradient formation is disrupted resulting in the loss of midline *Race* expression, and expansion of the *ush* expression domain (right hand cartoons). **B** (Top) Western blot showing the amount of Sog:Myc, Tsg:His and Tsg$^{I40E}$:His proteins co-immunoprecipitated with Dpp:HA–Scw:Flag heterodimers immobilized on anti-Flag matrix. Protein combinations are indicated in the key, Dpp:HA detection on the Flag matrix confirms immobilization of Dpp:HA–Scw:Flag heterodimers. (Bottom) Western blot shows the relative expression levels of wild-type and mutant Tsg proteins in the inputs. **C** Schematic of the CRISPR-Cas9 and homology-directed recombination strategy used to replace the *tsg* locus with an attP landing site and ΦC31-mediated reintegration of *tsg* rescue sequences. Dark blue regions indicate

genomic sequences excised along with *tsg* coding sequences (light blue). An ALFA epitope tag is added to the Tsg C-terminus. **D** smFISH analysis of the *Race* and *ush* expression domains in wild-type and *tsg* mutant embryos at the onset of gastrulation (dorsal views). Nuclei are stained with DAPI in blue. Scale bars: 50 µm. **E** *Race* and *ush* smFISH staining in wild-type, *tsg:ALFA* and *tsgI40:ALFA* mutant embryos at the onset of gastrulation. Nuclei are stained with DAPI (blue), scale bars: 50 µm. **F**, **G** Graphs show quantification of the number of nuclei from the dorsal midline in the center of the embryo expressing *Race* (**F**) or *ush* (**G**) based on the data shown in **D**, **E**. $n = 5$, biologically independent animals/embryos. Error bars represent standard deviations. **F** One-way ANOVA, Šídák's multiple comparisons test: ns, $P > 0.05$; ****, $P \leq 0.0001$ (wild-type vs. *tsg$^{-2}$* $P = 2.9 \times 10^{-14}$; wild-type vs *tsg$^{attP}$* $P = 2.9 \times 10^{-14}$; wild-type vs *tsg:ALFA* $P > 0.999$; *tsg:ALFA* vs *tsg$^{I40A}$:ALFA* $P = 2.9 \times 10^{-14}$; *tsg:ALFA* vs *tsg$^{I40E}$:ALFA* $P = 2.9 \times 10^{-14}$). **G** One-way ANOVA, Šídák's multiple comparisons test: ns, $P > 0.05$; ****, $P \leq 0.0001$ (wild-type vs. *tsg$^{-2}$* $P < 1.0 \times 10^{-15}$; wild-type vs *tsg$^{attP}$* $P < 1.0 \times 10^{-15}$; wild type vs *tsg:ALFA* $P > 0.999$; *tsg:ALFA* vs *tsg$^{I40A}$:ALFA* $P = 1.0 \times 10^{-15}$; *tsg:ALFA* vs *tsg$^{I40E}$:ALFA* $P = 2.0 \times 10^{-15}$).

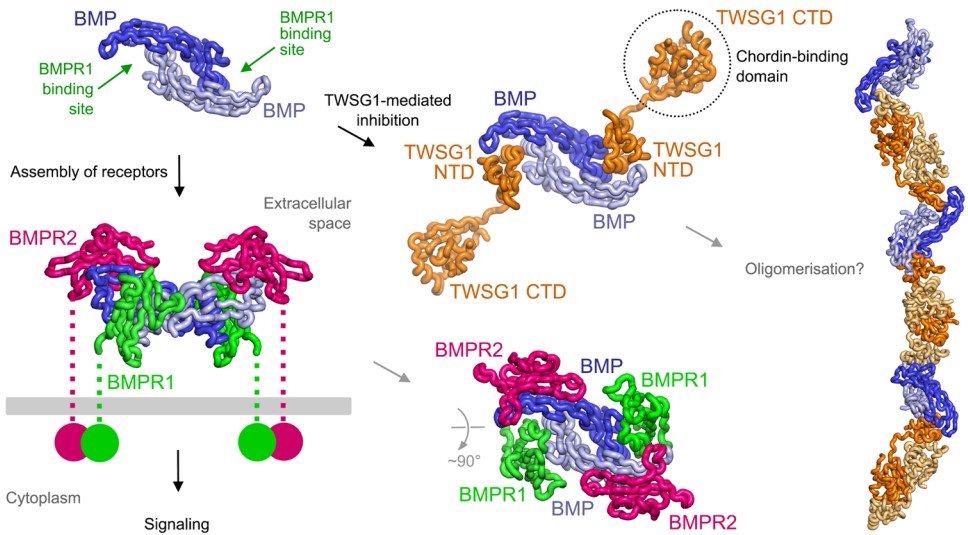

**Fig. 5 | Model illustrating the inhibition of BMP signaling by TWSG1.** Dimeric BMP ligands assemble a complex that comprises two BMPR1 and two BMPR2 receptors to activate downstream signaling. The TWSG1 NTD occupies the BMPR1-binding site on BMP ligands and inhibits signaling by competing with BMPR1 for binding. The TWSG1 CTD interacts with members of the Chordin family, adding

another layer to BMP–TWSG1 regulation. The dimeric nature of both the BMP ligand and TWSG1 (observed in structures and in solution) could lead to the oligomerization and clustering of the BMP–TWSG1 complex. Ternary signaling complexes were modeled based on the BMPR1A–BMP2–ActR2b structure (PDB ID 2H62)[24], as well as the TWSG1 apo and TWSG1–GDF5 structures (this study).

with BMPR1 binding. Indeed, TWSG1 readily inhibits BMP7 and GDF5 signaling in our cellular assays, and mutagenesis of interface residues impairs TWSG1 interactions with three BMP ligands, as well as abolishes TWSG1's BMP-inhibitory function in both cellular signaling and organoid-based assays (Fig. 3).

Serendipitously, the TWSG1–GDF5 complex crystallized in the presence of CaCl$_2$, and we identified a calcium-binding site that mediated TWSG1–GDF5 interactions at the periphery of the binding interface. Given that calcium ions are present in the extracellular space in millimolar concentration and thus could modulate BMP interactions with their binding partners, we investigated the role of calcium in both BMP interactions with TWSG1 and BMP type 1 receptors. The mutation of the calcium-binding residue Asp34 to Ala in TWSG1 did not abolish the TWSG1–BMP interactions, indicating a minor regulatory role for calcium. Similarly, the interactions between GDF5 and BMPR1B were weakened (though not completely abolished) in the presence of the calcium chelator EDTA. Since the calcium-binding residues in BMP ligands are conserved, and calcium is ubiquitously present in the extracellular space, our TWSG1–GDF5 structure brings this ion into focus. It is common to perform SPR-based binding assays in the presence of EDTA. Our findings highlight that the use of buffers

containing EDTA in BMP-binding studies should be considered with care. Notably, extracellular calcium promotes BMP2 signaling and bone formation from bone marrow stem cells[46], and a calcium gradient peaking at the dorsal midline in stage 5 *Drosophila* embryos is needed for amnioserosa formation[47].

TWSG1 can act as both an agonist and antagonist of BMP signaling[12-15]. Our molecular analyses support a mechanism of BMP antagonism by which TWSG1 competes with receptor binding (Fig. 5). However, it does not explain TWSG1's pro-BMP effects. A model for TWSG1's functions as a BMP-agonist is that it induces a conformational change in Chordin/Sog, rendering it more susceptible to cleavage by the protease Tolloid, thus promoting degradation of a BMP antagonist (Chordin/Sog) and boosting BMP signaling[13,14,30]. Although no direct interaction between Tsg and Tld has been detected[15,30], Tsg increases the rate of Chordin/Sog cleavage by Tld[15,17,48,49].

We show that the Chordin family member CHRDL2 directly binds to the TWSG1 CTD (but not NTD). Given that the BMP-binding NTD and the CHRDL2-binding CTD of TWSG1 are separated by an extended linker, it is possible that TWSG1 could bind to both BMP ligands and Chordin family members simultaneously. Our structural data also suggest a role for oligomerization of this complex, since a dimeric

GDF5 could bind to a dimeric TWSG1, which in turn could potentially bind to a monomeric or dimeric Chordin. Cell culture co-immunoprecipitation assays have identified both monomeric and dimeric *Xenopus* Tsg, as well as a species consistent with *Xenopus* BMP/TWSG1/Chordin complexes[13]. However, the stoichiometry of the ternary BMP–TWSG1–Chordin complex, how it affects the BMP-inhibitory function of Chordin and Chordin's interactions with Tolloid proteases remain to be elucidated.

Our data show that TWSG1–BMP interactions are evolutionarily conserved across species (Fig. 4). Tsg Ile40Ala/Glu mutations that disrupt BMP binding confer male lethality in *Drosophila*. The embryos show loss of pMad accumulation at the dorsal midline and disrupted BMP target gene expression that is typical of *sog* or *tsg* mutant embryos lacking a BMP gradient[14,37,42]. However, the expanded *Race* head spots in the anterior of the embryo are stronger in *tsg*[I40A/E] mutants compared to the *tsg*[2] and *tsg*[attP] mutants. It has been shown that the *Xenopus* Tsg Trp67Gly mutant, which cannot bind BMP, can still interact with Chordin[20]. As this Tsg mutant is strongly ventralizing (pro-BMP) in *Xenopus* embryos, it was proposed that Tsg Trp67Gly sequesters Chordin and prevents it from antagonizing BMP signaling[20]. We observed less Sog bound to Dpp–Scw in the presence of Tsg[I40E], compared to wild-type Tsg, in the S2 tissue culture assay, potentially because the Tsg[I40E] mutant sequesters Sog. Therefore, we speculate that the Tsg Ile40Ala/Glu mutants that we have characterized are able to bind Sog and prevent Sog-mediated BMP inhibition in vivo, resulting in stronger expression of the *Race* head spots. In contrast, in *tsg*[attP] embryos completely lacking Tsg, Sog would inhibit BMPs resulting in weaker *Race* expression. Further support for an inhibitory BMP–Sog complex being able to form in the absence of Tsg in vivo comes from the finding that *tsg* mutant embryos have lower BMP–receptor interactions and pMad levels than *tsg sog* double mutant embryos[44].

The role of TWSG1 in human diseases is emerging[50]. Previous studies demonstrated the antifibrogenic effect of BMP7 in kidney disorders[51,52]. TWSG1 is the major negative regulator of BMP7 in kidney podocytes and Twsg1-null mice are resistant to podocyte injury[18]. Thus, inhibition of TWSG1 function and boosting BMP7 signaling in podocytes is a potential therapeutic approach to treat kidney injuries. Similarly to the TWSG1–BMP7 signaling axis in the kidney, TWSG1 suppressed BMP7-enhanced migration of endometrial cancer cells, highlighting the tumor-inhibiting function of TWSG1[53]. We show that the NTD of TWSG1 binds to BMP7 with high affinity and inhibits BMP7 signaling in cellular assays (Figs. 1F and 3E). Our structural and functional data point to a specific region of TWSG1 (Ile40 of the NTD, Fig. 2B) that could be targeted with monoclonal antibodies or other therapeutic molecules in order to prevent TWSG1–BMP interactions and enhance BMP signaling in kidney injuries for the benefit of patients. On the other hand, our molecular analyses could be used to design TWSG1-based BMP inhibitors to interfere with specific BMP ligand–receptor interactions.

## Methods

### Cloning, expression, and purification of TWSG1 and CHRDL2 constructs

Cell lines and plasmids used in this study are listed in Supplementary Table 2. A synthetic gene (codons optimized for expression in mammalian cells, Thermo Fisher Scientific) encoding human TWSG1 (UniProt ID Q9GZX9) was cloned (using Age1 and Kpn1 restriction enzymes from New England Biolabs) into the pHLsec vector[54], resulting in an expression construct with an N-terminal secretion signal, followed by ETG, TWSG1 (residues C26–F223), and a C-terminal GTKH$_6$ tag. This construct was used to produce TWSG1 for cellular and SPR assays, but not crystallography. For crystallization, cDNA (Bioscience Gene Service)[30] encoding full-length human TWSG1 (UniProt ID Q9GZX9) was cloned (using Age1 and Kpn1 restriction enzymes from New England Biolabs) into the pHLsec vector[54], resulting in an expression

construct with an N-terminal secretion signal, followed by ETG, TWSG1 (residues C26–F223), and a C-terminal GTKH$_6$ tag. cDNA (Bioscience Gene Service)[30] encoding human TWSG1 NTD (C26–S83) was cloned into a modified pHR-CMV-TetO2 vector[55], resulting in an expression construct with an N-terminal secretion signal, followed by ETG, TWSG1$_{NTD}$, GTLEVLFQGP (HRV 3C protease recognition sequence), a linker (GGS)$_3$, mono Venus (M1–K239), and a C-terminal H$_6$SGSH$_6$ tag. cDNA (Bioscience Gene Service)[30] encoding human TWSG1 CTD (T85–F223) was cloned into a modified pHR-CMV-TetO2 vector[55], resulting in an expression construct with an N-terminal secretion signal, followed by ETG, TWSG1 NTD, and a C-terminal GTKH$_6$ tag. Human CHRDL2 (A25–T429, UniProt ID Q6WN34) was cloned into the pHLsec vector, resulting in an expression construct with an N-terminal secretion signal, followed by ETG, CHRDL2, and a C-terminal GTKH$_6$ tag.

Full-length human TWSG1 for crystallization was expressed by transient transfection in HEK-293T cells (ATCC CRL-11268) in the presence of the class I α-mannosidase inhibitor kifunensine for ~72 hours at 37 °C[54,55]. Media with secreted proteins were centrifuged (10,000 × *g*, 30 min, 21 °C), filtered (0.22-μm polyethersulfone membrane; Millipore), and dialyzed against 574 mM NaCl, 5.4 mM KCl, and 20 mM phosphate buffer (pH 7.4 at 25 °C) using a QuixStand benchtop diafiltration system (GE Healthcare) (~21 °C, ~6 hours). Proteins were purified using immobilized metal (cobalt) affinity chromatography (IMAC, TALON resin; Clontech). For crystallization, full-length TWSG1 was purified by SEC (typically in 150 mM NaCl, 10 mM HEPES pH 7.5). Peak fractions were pooled, concentrated and deglycosylated with endoglycosidase F1 (~10 μg per mg of target protein, ~1 hour at 21 °C) to cut the Asn-linked glycans down to one N-acetylglucosamine moiety, for crystallization (Supplementary Table 1).

TWSG1 NTD (fused to mono Venus-His$_{12}$) was expressed and purified from dialyzed media using IMAC similar to full-length TWSG1. Following IMAC, TWSG1 NTD was dialyzed against 150 mM NaCl, 10 mM HEPES pH 7.5, 1% (v/v) glycerol, 0.02% (w/v) NaN$_3$ for 5 hours at 21 °C, then 18 hours at 4 °C using 2 kDa molecular weight cut-off Slide-A-Lyzer dialysis cassettes (Thermo Fisher Scientific). Dialyzed protein (~1.1 mg/ml, ~110 ml) was supplemented with endoglycosidase F1 (1 mg/ml, 0.2 ml), His-tagged HRV 3 C protease (2.5 mg/ml, 0.2 ml), and further dialyzed against 150 mM NaCl, 10 mM HEPES pH 7.5, 1% (v/v) glycerol, 0.02% (w/v) NaN$_3$ for 18 hours at 21 °C. Because the TWSG1 NTD-mono Venus fusion protein was not fully cleaved, additional 3 C protease was added (2.5 mg/ml, 0.3 ml) and proteins were incubated for additional 2 days at 4 °C. His-tagged mono Venus and 3 C protease was removed from TWSG1 NTD using IMAC. The remaining TWSG1 NTD was concentrated and purified using SEC in 1 M NaCl, 0.1 M Tris pH 8.0, 0.02% NaN$_3$ (HiLoad 16/60 Superdex 75 column; GE Healthcare, 4 °C). SEC fractions containing TWSG1 NTD were pooled, concentrated to 10.3 mg/ml, and stored at −80 °C.

TWSG1 CTD and CHRDL2 were expressed and purified from dialyzed media using IMAC followed by SEC as the full-length TWSG1. Proteins for cellular and binding assays, SEC-MALS experiments were produced without kifunensine and kept fully glycosylated. Site-directed mutagenesis of TWSG1 was performed by a two-step overlap extension PCR and constructs were verified by DNA sequencing. Purified proteins were analyzed using SDS-PAGE (Supplementary Fig. 12).

### Expression and purification of GDF5

A synthetic gene (codons optimized for expression in *E.coli*, Thermo Fisher Scientific) encoding human GDF5 (A382–R501, UniProt ID P43026) was cloned (using Nde1 and Xho1 restriction enzymes from New England Biolabs) into the pET-22b(+) vector (Novagen), resulting in an expression construct with an N-terminal methionine, followed by GDF5 without any tags. GDF5 was expressed in *E. coli* Rosetta(DE3) pLysS cells (Novagen). Bacterial cultures were grown at 37 °C (240 rpm, 2 liter flasks, 0.5 liter culture/flask) to an optical density of 0.8 (at 600 nm). GDF5 expression was induced with 1 mM isopropyl-

beta-D-thiogalactopyranoside (IPTG) and then grown for ~5 hours before harvesting. Cells were disrupted by sonication, and GDF5-containing inclusion bodies were washed in 800 mM NaCl, 20 mM Tris pH 8.0, 20 mM EDTA pH 8.0, 2% (v/v) Triton X-100. GDF5 inclusion bodies were solubilized in 8 M urea, 50 mM Na acetate pH 5, 1 mM DTT. Refolding was started by diluting GDF5 (14 mg/ml, 1 ml) into 100 ml of 0.1 M Tris, 10 mM EDTA, 1 M NaCl, 1% (w/v) CHAPS, 5 mM reduced L-glutathione, 2.5 mM oxidized L-glutathione, pH adjusted to 8.8, and performed for 72 hours at 4 °C. Refolded GDF5 was concentrated using and purified using SEC in 1 M NaCl, 0.1 M Tris pH 8.0, 0.02% NaN$_3$ (HiLoad 16/60 Superdex 75 column; GE Healthcare, 4 °C). SEC fractions containing GDF5 were pooled, concentrated to 9–14 mg/ml, and stored at −80 °C.

### Expression and purification of Gremlin-1

The cDNA (IMAGE clone 7262108) encoding human Gremlin-1 (V73–D184, UniProt ID O60565) was cloned (using Nde1 and Xho1 restriction enzymes from New England Biolabs) into the pET-22b(+) vector (Novagen), resulting in an expression construct with an N-terminal methionine, followed by Gremlin-1 without any tags. Gremlin-1 was expressed in *E. coli* Rosetta(DE3)pLysS cells (Novagen). Bacterial cultures were grown at 37 °C (180 rpm, 2 liter flasks, 0.5 liter culture/flask) to an optical density of 0.8 (at 600 nm). Gremlin-1 expression was induced with 1 mM IPTG and then grown for 6 hours before harvesting. Cells were disrupted by sonication and Gremlin-1-containing inclusion bodies were washed in 800 mM NaCl, 25 mM Tris pH 8.0, 20 mM EDTA pH 8.0, 2% (v/w) Triton X-100. Gremlin-1 inclusion bodies were solubilized in 6 M guanidinium chloride, 0.1 M Tris pH 8.0, 1 mM EDTA pH 8.0, 0.1 M DTT and then dialyzed against 5 M guanidinium chloride, pH ~3 for ~18 hours at 4 °C using 7 kDa molecular weight cut-off Slide-A-Lyzer dialysis cassettes (Thermo Fisher Scientific). Refolding was started by diluting Gremlin-1 (~7.6 mg/ml, ~10 ml) into 400 ml of 0.1 M Tris, 5 mM EDTA, 1 M L-Arg, 0.1 mM reduced L-glutathione, 0.1 mM oxidized L-glutathione, final pH adjusted to 8.3, and performed for ~18 hours at 4 °C. Refolded protein (~20 ml, 3.5 mg/ml) was dialyzed against 20 mM HEPES pH 8.0, 5% glycerol for ~18 hours at 4 °C. Gremlin-1 precipitated after the dialysis. Precipitated Gremlin-1 was solubilized in 6 ml of 0.5 M L-Arg, 0.4 M NaCl, 5% glycerol, final pH 8.0, and purified using SEC in PBS buffer supplemented with 0.4 M NaCl, 5% glycerol (HiLoad 16/60 Superdex 200 column; GE Healthcare, 21 °C). SEC fractions with Gremlin-1 were pooled, loaded onto a heparin column (5 ml HiTrap HP from GE Healthcare, 21 °C), and eluted with a NaCl gradient (linear gradient from PBS, 5% glycerol, 0.4 M NaCl to PBS, 1 M NaCl, 5% glycerol). Fractions containing Gremlin-1 were pooled, dialyzed against PBS buffer supplemented with 0.4 M NaCl, 5% glycerol (~18 hours at 4 °C), concentrated to 0.8 mg/ml, and stored at −80 °C.

### Crystallization of the full-length TWSG1 and the TWSG1 NTD−GDF5 binary complex

Details about crystallization and cryoprotection conditions are presented in Supplementary Table 1. For the TWSG1 NTD−GDF5, GDF5 (14.11 mg/ml, 0.213 ml) and TWSG1 NTD (10.28 mg/ml, 0.159 ml) were mixed 1:1 mol:mol (both proteins were in 1 M NaCl, 0.1 M Tris pH 8, 0.02% NaN$_3$) and dialyzed against 150 mM NaCl, 10 mM HEPES pH 7.5, 0.02% NaN$_3$ for ~18 hours at 4 °C. The protein complex was concentrated to 20 mg/ml and crystallized using sitting drop nanoliter vapor diffusion[56]. Initial crystals were obtained using the Morpheus crystallization screen (wells A8 and A12)[57] and further optimized to include CaCl$_2$ but not MgCl$_2$ (Supplementary Table 1).

### X-ray data collection, structure determination, and refinement of TWSG1 and the TWSG1−GDF5 complex

Data were collected at the Diamond Light Source UK (DLS), indexed, and integrated using XDS[58], scaled using AIMLESS[59] in xia2[60]

(Supplementary Table 1). Initial phases of X-ray diffraction reflections were determined using data from native TWSG1 crystals ($d_{min}$ = 2.6 Å, space group P6$_5$22, 2 molecules/asymmetric unit, crystal form 1) and single-wavelength anomalous diffraction data ($d_{min}$ = 3.6 Å, P6$_5$22, crystal form 1) from crystals soaked in di-μ-iodobis(ethylenediamine) diplatinum(II) nitrate (PIP) (saturated solution, 2 h at 21 °C) using autoSHARP[61] (Supplementary Table 1). The initial model was built using Phenix AutoBuild[62], further refined using Coot[63] and Phenix[64]. Anisotropic data (4 out 6 datasets, Supplementary Table 1) were corrected using STARANISO (Global Phasing Ltd.) implemented in the DLS processing pipeline. The TWSG1−GDF5 complex was solved by molecular replacement using GDF5 (PDB ID 6Z3J[11]) and TWSG1$_{NTD}$ as search models in Phaser[65], refined using Coot and Phenix. The refined TWSG1−GDF5 complex at 1.96 Å was used to phase two long-wavelength X-ray datasets (Supplementary Table 1). Phased anomalous difference Fourier maps were calculated using the phenix.find_peaks_holes program in Phenix. Anomalous scattering factors from X-ray fluorescence data were derived using CHOOCH[66]. The quality of structures was evaluated using Coot[63] and MolProbity[67]. Snapshots of electron density maps are presented in Supplementary Fig. 13. Figures of structures were prepared using PyMOL (The PyMOL Molecular Graphics System, version 2.3.2, Schrödinger).

### SPR-based binding studies

SPR experiments were performed using a Biacore T200 instrument (GE Healthcare) at 25 °C. Ligands were covalently linked to the surface of series S CM5 chip (Cytiva) via primary amine coupling. BMP2[68], BMP7 (Miltenyi Biotec), and GDF5 (this study) were diluted to 0.04 mg/ml in 10 mM Na acetate pH 4.0 for amine coupling. Running buffer was 0.15 M NaCl, 20 mM HEPES pH 7.5, 2 mM CaCl$_2$ (or 2 mM EDTA pH 8.0, Supplementary Fig. 5C, D, G, H, K, L), 0.005% Tween 20 (flow rate 10 μl/min). All analytes were purified by SEC in SPR running buffer before use, and 1:2 dilution series were prepared. The ligand-analyte interaction time was set to 15 min. After each interaction measurement and dissociation, the chip surface was further regenerated with 4 M MgCl$_2$ (100 μl/min, 3 min). The signal from experimental flow cells was corrected by subtraction of a buffer and reference signal from a flow cell without coupled protein. Equilibrium dissociation constants ($K_d$s) and maximum analyte binding (B$_{max}$) were calculated using the GraphPad Prism 9 by fitting data to a 1:1 binding isotherm model: $y = (B_{max} \times x)/(K_d + x)$; where $y$ is binding response, $x$ is the analyte concentration. Analyte concentrations were determined from the absorbance at 280 nm using a NanoDrop ND-1000 spectrophotometer (Thermo Fisher Scientific) and the calculated molar extinction coefficients from the ProtParam webserver (https://web.expasy.org/protparam/). We acknowledge that coupling BMPs to the SPR chip via primary amines might influence their interactions with analytes, potentially leading to $K_d$s derived from our experiments differing from those observed in vivo.

### Cellular assays of BMP signaling in C2C12 cells

C2C12 cells (stably transfected with a reporter plasmid consisting of the BMP response element (BRE) from the Id1 promoter fused to a luciferase reporter gene[27], were grown in DMEM high-glucose media (Sigma) supplemented with 2 mM L-glutamine (Gibco), non-essential amino acids (Gibco), and 10% FBS (Gibco) at 37 °C, 5% CO$_2$. For assays, cells were plated in complete DMEM, 10% FBS at a density of 50,000 cells per well (100 μl per well) in a 96-well plate (Nunc-Immuno MicroWell polystyrene plates with Nunclon delta surface, Sigma). Media were changed to complete DMEM, 0.1% FBS after 24 hours. To trigger signaling, media were changed to complete DMEM, 0.1% FBS supplemented with purified BMP ligands with or without TWSG1 (or Gremlin-1). After 48 hour incubation with BMP ligands and their binders, cells were lysed, and the activity of firefly luciferase was measured using the Dual-Glo luciferase assay system (Promega). Luminescence was measured using a Luminoskan Ascent luminometer (Labsystems).

## C2C12 pSmad assay

C2C12 cells were plated on glass coverslips (no.1, Knittel Glass, VD10013Y1B.01) at a density of 100,000 cells per well in a 24-well plate in DMEM/F-12 Ham (Sigma) supplemented with 1% penicillin/streptomycin, 10% FBS and cultured at 37 °C, 5% $CO_2$. Cells were incubated in serum-free media overnight. To induce signaling, media was changed to serum-free media supplemented with BMP/GDF ligands (10 nM BMP7, 10 nM BMP2, 40 nM GDF5) with or without TWSG1/TWSG1 Ile40Glu for 1 hour.

## C2C12 cellular differentiation assay

C2C12 cells were plated on glass coverslips as described above, but at a density of 30,000 cells per well. After 24 hours when mostly confluent, media was changed to DMEM/F-12 Ham (Sigma) supplemented with 1% penicillin/streptomycin, 1% FBS with BMP7, with and without TWSG1/ TWSG1 Ile40Glu, and cultured for a further 72 hours.

## C2C12 immunofluorescence

For immunofluorescence, C2C12 cells were washed in PBS and fixed with 4% PFA for 15 mins, quenched and permeabilised in 0.2 M glycine, 0.5% Triton for 30 mins and blocked with 2% fish skin gelatine. Samples were stained with primary antibody for 1 hour, washed in PBS before adding secondary antibody for 1 hour. Then cells were washed in PBS and water before mounting in ProLong Gold antifade reagent with DAPI (Thermo Fisher Scientific P36935). All steps were performed at room temperature.

pSmad was visualized by staining with a rabbit anti-Smad3 (phospho S423 + 425) PUR) (1:750 dilution) primary antibody that cross-reacts with phospho-Smad1[69] and a donkey anti-rabbit IgG Alexa Fluor 647 (1:500) secondary antibody. For cellular differentiation assays, samples were stained for Alkaline Phosphatase with a rabbit anti-ALPL (1:500) primary antibody and Myosin Heavy Chain IV with a mouse anti-MHC IV Alexa Fluor 488 conjugated (1:100) primary antibody and donkey anti-rabbit IgG Alexa Fluor 647 (1:500) secondary antibody. Experiments were performed in triplicate, antibody catalog numbers are listed in Supplementary Table 2.

## C2C12 imaging

For pSmad immunofluorescence, samples were imaged on a Leica TCS SP8 AOBS confocal microscope with a HC PL APO CS2 ×40/1.30 oil objective. Images were collected at 1× zoom, pinhole 1 airy unit, scan speed 600 Hz bidirectional, 2048 × 2048 format, at 8-bit, 3× Line Averaging, 0.5 μm Z step size, using the white light laser with 405 nm (5%) and 647 nm (10%) and hybrid detectors. Images were processed in Fiji ImageJ[70]. To quantify the number of pSmad active cells, nuclei were counted in the 405 nm channel and pSmad active nuclei counted in the 647 nm channel in Fiji ImageJ. Briefly, a gaussian blur was applied before thresholding pSmad intensity based on the blank sample using a Li algorithm, the image converted to a binary and a watershed method applied to split overlapping objects which were counted using the analyze particles tool. Statistical analyses were performed in GraphPad Prism 10 (10.0.3).

For C2C12 differentiation assays, samples were imaged on a Leica TCS SP8 AOBS confocal microscope with a HC PL APO CS2 40×/1.30 oil objective. Images were collected at 1× zoom, pinhole 1 airy unit, scan speed 600 Hz bidirectional, 2048 × 2048 format, at 8-bit, 3× Line Averaging, 0.35 μm Z step size, using the white light laser with 405 nm (5%), 488 nm (10%) and 647 nm (10%) and hybrid detectors. Raw images were deconvolved with Huygens Professional software (v 23.04) (SVI) and processed in Fiji ImageJ.

## SEC coupled to multi-angle light scattering (SEC-MALS)

SEC-MALS experiments were performed using a Wyatt Dawn HELEOS-II 8-angle light-scattering detector (with 663.8-nm laser) and a Wyatt Optilab rEX refractive index monitor linked to a Shimadzu HPLC system comprising LC-20AD pump, SIL-20A autosampler, and SPD20A UV/Vis detector. SEC-MALS of proteins (2 mg/ml, 0.1 ml per injection) was performed using a Superdex 200 HR 10/30 column equilibrated in 150 mM NaCl, 10 mM HEPES pH 7.5, 2 mM $CaCl_2$ at 0.5 ml/min flow rate and 21 °C. Scattering data were analyzed, and the molecular mass was calculated using ASTRA 6 software (Wyatt Technology). dn/dc values 0.185 and 0.146 ml/g were used for proteins and glycans, respectively[71,72]. It was assumed that each Asn-linked $Man_9GlcNAc_2$ glycan contributed 1.883 kDa to protein size. Predicted glycosylated Asn residues: 52, 81, and 147 in human TWSG1; 114 in human CHRDL2.

## Mice

All procedures were carried out in accordance with UK Home Office regulations and the Animals (Scientific Procedures) Act 1986. All mice were housed in individually ventilated cages at the animal unit at the Functional Genetics Facility (Wellcome Center for Human Genetics, University of Oxford). They were housed in a specific pathogen-free (SPF) facility with unrestricted access to food and water and had not been involved in any previous procedures. All strains were maintained on a C57BL/6 J background for R6 generations. Procedures were conducted on mice at least 6 weeks of age, including both males and females.

## Intestinal organoid-based assay of TWSG1 function

Mouse intestinal crypts were isolated and cultured as described previously[34]. In brief, crypts were isolated, resuspended in Matrigel (BD Biosciences), and plated out in 24-well plates in 20 μl domes. The basal culture medium (advanced Dulbecco's modified Eagle medium/F-12 supplemented with penicillin/streptomycin, 10 mM HEPES pH 7.4, Glutamax, 1× N2, 1× B27 (all from Invitrogen), and 1 mM N-acetylcysteine (Sigma)) was overlaid containing the following growth factors: Epidermal Growth Factor at 50 ng/ml (Life Technologies), Noggin at 100 ng/ml (PeproTech) and R-spondin-1 at 500 ng/ml (R&D Systems) (ENR media). The media were changed every 2–3 days. To test TSWG1 as a BMP antagonist, Noggin was substituted with wild-type or mutant TWSG1 at 10 μg/ml (425 nM). Organoids were passaged three times in order to allow a sufficient period for observable phenotype development. Passage protocol consisted of rigorous shear mechanical disruption consisting of vigorous pipetting of media-suspended organoids with a 1 ml pipette 50 times. Organoids in each condition were pelleted via centrifugation at $300 \times g$ at 4 °C for 5 minutes, before resuspension in Matrigel and re-plating. In order to assess organoid persistence and survival phenotype, organoids in all experimental conditions were plated at an equivalent confluence upon the first passage, and there was no discard of cellular material across passage cycles. Organoids were imaged at ×10 magnification using the EVOS M5000 imaging system (Invitrogen). For assessment of organoid survival, the full depth of the Matrigel dome was imaged using an Olympus SpinSR SoRa system at ×4 magnification. Each well was imaged as a Z-stack of 15 levels collected over a depth of 1200 μm, to capture all organoids suspended within the Matrigel, and quantified via manual annotation in QuPath image analysis software.

## RT-qPCR of Bmp transcript expression in mouse intestinal organoids

Organoids were collected 3 days post-passage. RNA was extracted using the RNeasy Mini Kit from Qiagen (74104), and DNase treatment was performed using the DNA-free kit from Life Technologies (AM1906) according to manufacturer instructions. Complementary DNA was obtained via reverse transcription PCR using the High Capacity cDNA Reverse Transcription Kit (Applied Biosystems). In the absence of an appropriate reference group, raw CT value was qualitatively reported to indicate relative expression of transcripts of interest. A higher CT value indicates lower expression, whereas CT lower than 30 suggests notable gene expression.

## Fly stocks

Fly stocks were maintained on a standard fly food media, yeast 50 g/l, glucose 78 g/l, maize 72 g/l, agar 8 g/l, 10% nipagen in ethanol 27 ml/l and propionic acid 3 ml/l. Fly stocks used in this work are listed in Supplementary Table 2. All *tsg* stocks were balanced with *FM7c-ftz-lacZ* to allow visualization of mutant embryos based on lack of *lacZ* staining. *y¹w⁶⁷c²³* flies were used as wild-type.

## Generation of the *tsg ᵃᵗᵗᴾ* fly stock

The *tsg ᵃᵗᵗᴾ* line was generated in *Drosophila melanogaster* using a two-step CRISPR-Cas9 with HDR strategy[38,39]. The CRISPR OptimalTarget Finder tool was used to identify PAM sites located up and downstream of the *tsg* locus and guide RNA sequences designed 3 nucleotides upstream for the creation of targeted double-stranded breaks at sites 3′ (dm6.34 X:11987851 (+strand)) and 5′ (dm6.34 X:11989264 (+strand)) of the *tsg* locus[73]. A guanosine was added to the 5′ end of the guide sequence for efficient expression under the dU6-2 promoter. Phosphorylated guide sequences (Sigma Aldrich) were cloned into the pU6-Bbs1-gRNA plasmid (Addgene 45946) using the Bbs1 restriction site as described[39,73]. Homology arms to these cut sites were then designed (dm6.34 3′ X: 11986861–11987852 (+strand) and 5′ X: 11989266–11990224 (+strand)), generated by PCR from genomic DNA and cloned into the pHD-DsRed-attP (Addgene 51019[73]) donor plasmid using Nde1 and Spe1 restriction sites and In-fusion cloning (Takara Biosciences). Donor plasmid and gRNA vectors were injected into embryos expressing Cas9 under a *nanos* promoter (BL78782) at the University of Manchester Fly Facility. Survivors were crossed to *y¹w⁶⁷c²* adults to screen for successful CRISPR events before balancing using *brkM68/FM7c-ftz-lacZ*[39,74]. The *dsRed* marker was subsequently removed by crossing *tsg ᵃᵗᵗᴾ/FM7c-ftz-lacZ* females to *FM7c-ftz-lacZ* balanced males carrying Cre recombinase on the third chromosome.

Reintegration plasmids were generated from the RIV^white plasmid, carrying the *mini-white* marker (gift from the Vincent lab[38]). Briefly, the sequence excised 5′ (X:11988914–11989265) and 3′ (X:11987854–119988162) to the *tsg* coding sequence was inserted into RIV^white between the attB and loxP sites using Kpn1 and EcoR1 restriction sites and In-fusion cloning. This RIV^white-template plasmid was subsequently PCR linearized, and the *tsg* CDS and ALFA-tag inserted between the 5′ and 3′ excised regions, creating a seamless sequence of the excised *tsg* locus, with the addition of a short epitope (ALFA) tag[75]. To generate point mutant reintegration vectors, *tsg* Ile40 was mutated to Ala (Ile40Ala) or Glu (Ile40Glu) using In-fusion cloning.

Reintegration plasmids were co-injected with a ΦC31 encoding plasmid into *tsg ᵃᵗᵗᴾ* embryos, and survivors crossed back to the balanced *tsg ᵃᵗᵗᴾ* line. Successful recombinants were screened using the *mini-white* marker and confirmed by sequencing genomic DNA.

## smFISH, immunostaining, and imaging of *Drosophila* embryos

Fixed embryos (2–4 hours) were processed as previously described[76]. For smFISH, samples were stained with *ush* Stellaris, *lacZ* Stellaris, and *Race* smiFISH probes[77]. For pMad immunostaining, embryos were stained with a *lacZ*-digoxygenin-UTP probe[77] followed by rabbit anti-Smad3 [(phospho S423 + 425)](1:750), which cross-reacts with pMad[78], and sheep anti-Digoxigenin [Fab fragments antibody, AP conjugated] (1:200) primary antibodies. Donkey anti-rabbit IgG Alexa Fluor 647 (1:500) and donkey anti-sheep IgG Alexa Fluor 488 (1:500) secondary antibodies were used. Nuclei were stained with DAPI (1:1000, NEB 4083) and mounted in ProLong™ Diamond Antifade Mountant (Thermo Fisher Scientific P36961). Details of the antibodies used for imaging are listed in Supplementary Table 2.

Samples were imaged on a Leica TCS SP8 AOBS confocal microscope with an HC PL APO CS2 ×40/1.30 oil objective. For smFISH, images were collected at 0.75 zoom, pinhole 1 airy unit, scan speed 600 Hz bidirectional, 2048 × 2048 format, at 8-bit, 4x Line Averaging,

0.35 μm Z step size, using the white light laser with 405 nm (5%), 548 nm (20%) and 647 nm (20%) and hybrid detectors. For immuno-fluorescence, images were collected at 0.75 zoom, pinhole 1 airy unit, scan speed 600 Hz bidirectional, 2048 × 2048 format, at 8-bit, 3x Line Averaging, 0.35 μm Z step size, using the white light laser with 405 nm (5%), 488 nm (10%) and 650 nm (10%) and hybrid detectors. Raw images were deconvolved with Huygens Remote Manager software v 3.7.1 (SVI). *tsg²*, *tsg ᵃᵗᵗᴾ*, *tsg ᴵ⁴⁰ᴬ*, and *tsg ᴵ⁴⁰ᴱ* hemizygous males were identified by the absence of a *lacZ* signal.

Deconvolved images were processed and analyzed in Fiji ImageJ[70]. To quantify the *Race* and *ush* expression domains, maximum intensity projections of the imaging stacks were made, the expression domain midline determined, and nuclei counted perpendicular to the midline at 50% embryo length. Statistical analysis was performed in GraphPad Prism 9 (v.9.4.1).

## S2 cells co-immunoprecipitation assay

S2R+ cells were cultured in Schneider's *Drosophila* media (Gibco) supplemented with 10% FBS, 1% penicillin/streptomycin at 25 °C. To produce Dpp-Scw heterodimers, cells were transfected as in ref. 37. Briefly, cells were transfected with 10 μg pMT-Dpp:HA and 4 μg pMT-Scw:Flag in a T75 flask (Corning). A pMT-BiP-Tsg^I40E:His expression plasmid was produced by introducing a single Ile40Glu point mutation to pMT-BiP-Tsg:His[79] by In-fusion cloning (Takara). To produce Sog, Tsg or Tsg^I40E, cells were transfected with 1 μg pMT-Sog:Myc, pMT-BiP-Tsg:His or pMT-BiP-Tsg^I40E:His in a 12-well plate. Protein expression was induced 24 hours after transfection with 0.5 mM CuSO₄ and media collected after a further 72 hours. Conditioned media from transfected cells was mixed and incubated at room temperature for 3 hours before binding to anti-Flag M2 matrix (Sigma) at 4 °C overnight. Matrix was subsequently collected and washed three times with buffer (20 mM Tris, pH 8, 150 mM NaCl) and samples eluted by boiling in NuPAGE LDS Sample Buffer (ThermoFisher Scientific) + 20% β-mercaptoethanol. Samples were analyzed by Western blot using rabbit anti-His (1:1000), mouse anti-Myc (1:1000) and chicken anti-HA (1:500) primary antibodies, with IRDye® 680RD donkey anti-Rabbit IgG (1:10,000), IRDye 800CW donkey anti-Mouse IgG (1:10,000) and IRDye 800CW donkey anti-Chicken IgG (1:5000) secondary antibodies. Antibody details are listed in Supplementary Table 2. Blots were visualized on a LI-COR Odyssey CLx. Samples were run in parallel blots to probe for both Sog:Myc and Dpp:HA due to antibody light-chain cross-reactivity obscuring the Dpp:HA band ~25 kDa.

## Statistics and reproducibility

Experiments with mouse colon organoids presented in Fig. 3 were performed at least twice by two different people (H.E. and H.L.B.-D.) with similar results. Experiments corresponding to Fig. 4B were performed in triplicate. Experiments corresponding to Supplementary Fig. 12 were performed once. Replicates of SPR-based binding experiments and cellular signaling assays are provided in the Supplementary Information file. *P* values are provided in the Source Data file.

## Reporting summary

Further information on research design is available in the Nature Portfolio Reporting Summary linked to this article.

# Data availability

Coordinates and structure factors have been deposited in the Protein Data Bank with the following accession numbers: 8BWA (TWSG1 + PIP, crystal form 1), 8BWD (TWSG1, crystal form 1), 8BWI (TWSG1, crystal form 2), 8BWL (GDF5 + TWSG1+calcium, native), 8BWM (GDF5 + TWSG1+calcium, 4042 eV), and 8BWN (GDF5 + TWSG1+calcium, 4010 eV). All other data required to evaluate the conclusions in the paper are present in the paper, the Supplementary Information and Source Data file. Source data are provided with this paper.

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

## Acknowledgements

We thank the staff of beamlines I03, I23, and I24 at the Diamond Light Source, UK, for assistance, and Thomas Walter, Karl Harlos, and David Staunton for technical support. Work was supported by Cancer Research UK (C20724/A26752 and DRCRPG-May23/100002 to C.S.) and the Biotechnology and Biological Sciences Research Council (BB/V008099/1 to H.L.A. and C.B). T.M. was supported by a long-term postdoctoral fellowship from the Human Frontier Science Program (LT000021/2014-L). R.E.W. was supported by a Wellcome Trust DPhil studentship (203726/Z/16/Z), and G.M. was supported by a BBSRC DTP PhD studentship (BB/M011208/1). The Wellcome Center for Human Genetics, Oxford, is funded by Wellcome Trust Core Award 090532/Z/09/Z, and the Wellcome Center for Cell-Matrix Research is funded by Wellcome Trust Core Award 203128/Z/16/Z. S.J.L. was supported by Cancer Research UK Program Grant (DRCNPG-Jun22\100002). We thank Thomas Mueller (University of Würzburg) for providing recombinantly expressed BMP2, Catherine Sutcliffe, Sanjai Patel, and the Manchester Fly Facility for assistance in generating the *tsg* *attP* fly line, Osamu Shimmi for advice on the S2 cell assay, and Steven Woods for advice on the C2C12 pSmad assay.

## Author contributions

T.M., C.B., H.L.A., and C.S. conceived the study. T.M. and A.F.R. expressed and purified proteins for functional studies and crystallization. T.M. and A.F.R. performed SEC-MALS and SPR experiments and crystallized proteins. T.M. carried out cellular reporter assays. T.M., A.F.R., and C.S. collected and processed X-ray diffraction data. T.M. and C.S. determined the structures and refined atomic models. H.E., H.L.B.-D., and S.J.L. performed the mouse organoid experiments. G.M. carried out the cellular pSmad immunofluorescence and differentation assays, and performed the fly and S2 cell experiments, supervised by H.L.A. and C.B. S.C.G and R.E.W. provided vectors for protein expression. K.E.O., R.D., and A.W. collected long-wavelength X-ray diffraction data. T.M., G.M., A.F.R., H.L.A., and C.S. wrote the paper with input from all authors.

## Competing interests

The authors declare no competing interests.
