## [Peer Review File · Nature Communications]

Molecular Mechanism of BMP Signal Control by Twisted GastrulationReviewer #1 (Remarks to the Author):

The manuscript by Malinauskas, Moore and Rudolf et al., describes a new structure of the BMP modulator twisted gastrulation (TWSG1). This is a wonderful study which sheds new light into the molecular binding mechanism of TWSG1 and how it interacts with BMP ligands. The authors show that the N-terminal cys rich domain adopts a helical structure which inserts a helix into the dimer interface to out-compete type I receptor binding. The C-terminus was shown to bind another BMP modulator of the chordin family. Collectively, this information starts to explain how TWSG1, chordin and BMP form an inhibitory complex, however, much is still to be determined and will be a focus of future efforts such as a ternary complex or a complex of the CTD with CHL2. The authors identify a key residue (I40) in the NTD that when mutated abolishes binding. A series of in vivo and in vitro cell assays nicely support the findings. That being said, I have divided the critiques into major and minor comments/concerns which should be addressed by the team of investigators. Overall, I am very positive for the publication of the study depending on how the concerns are addressed.

Major points:

1. There is a disconnect with the use of the truncated vs full-length TWSG1. Binding experiments were performed with a truncated version and the activity assays were performed with the full-length. How does full-length binding compare to the individual domains and also does the NTD of TWSG1 inhibit BMP signaling to the full potential as TWSG1? While the CTD alone does not bind in SPR experiments it may contribute to binding in the full-length version.
2. SPR-
 - a. SPR traces need to be added for all equilibrium experiments, especially for BMP2. Some were provided in sup figure 2 and some have odd behavior such as panel F.
 - b. There are significant concerns over the SPR experiments that would take considerable effort to resolve. One limitation of the SPR is that primary amine coupling could be affecting some of the ligands differently. This is certainly the case where the different ligands have a different number of lysine residues. Ideally, one would flip the orientation of the experiment and test ligand binding to a tagged version of the different TWSG1. Also, it appears the SPR experiments were only performed a single time with a single (very high) coupling density. A more appropriate experiment would have been to couple a CM5 chip at different densities and perform the equilibrium binding experiment at least twice. In the interest of time and getting this to the public I think the authors need to, at the minimum, provide a replicate for the binding data and include the full length TWSG1 for comparison.
3. A fair bit of the story includes a Ca binding site for GDF5 (and other ligands) and how Ca enhances binding affinity of TSWG1 to the ligand. Receptor-ligand interactions were tested but not much of an impact was noted. While this might be important, experiments beyond the binding experiments were not included to support the claim that Ca plays a role in the function of TWSG1 or BMP ligands. First, can the authors comment on if a Ca appears in the high-resolution structures of other ligands or complexes currently published (perhaps modeled as a water but has similar coordination). Secondly, since it appears that Ca binding of the NTD of TWSG1 hinges on aspartate 34, the authors should examine binding of the NTD with D34 to an alanine. Alternatively, one could mutate D416 of GDF5 and test binding. As a second alternative, the authors might exclude the Ca part of the story which might be better for a follow-up study where there is more room to developed and test the function of Ca in BMP ligands and binding to modulators.
4. The specificity of TWSG1 for GDF5 and BMP7 and not BMP2 is interesting. The authors claim that TWSG1 does not inhibit BMP2 (and presumably BMP4), however, a BMP-like inhibition is required in the intestinal organoid assay. The authors use BMP inhibition generically to describe this assay. What BMP is present that needs to be inhibited for the organoid to form? Isn't it BMP4? If so, can the authors reconcile the specificity of TWSG1. Also, can the authors suggest a structural explanation for why TWSG1 binds more favorable to GDF5 and BMP7 than BMP2?
5. The exciting experiment where the authors compare the IC50 of BMP signaling with CHL2 alone, TWSG1 alone, CHL2+TWSG1, CHL2+TWSG1 (NTD), CHL2+TWSG1 (CTD) is missing. Since the authors have all the components and cell-based assays this would be straightforward and tie into the data where the authors shown the CTD binds CHL2.

Minor points:

- A gel showing the quality of all in-house proteins produced and purified should be included (both reduced and non-reduced) .
- (Intro) While sclerostin was originally described as a BMP antagonist, it only inhibits Wnt signaling and this is the mode of action for the Amgen drug, Romosozumab.
- (Figure 2) Finger 1 and Finger 2 of the ligand are mislabeled.
- Why was EDTA and not EGTA used for the Ca chelating experiments?

Reviewer #2 (Remarks to the Author):

This paper describes the crystal structure of Twisted gastrulation (TWSG1), a conserved BMP binding protein. The authors also investigate the crystal structure of a complex of TWSG1 and GDF5. These studies reveal that the N-terminal domain of TWSG1 forms a complex with BMP ligands, while the C-terminal domain mediates binding with Chordin-like proteins. This study further identifies Ile40 as a key amino acid residue for TWSG1 and BMP ligand interactions. The authors then investigate the protein interactions between TWSG1 and BMP ligands using surface plasmon resonance (SPR)-based equilibrium binding experiments. These approaches confirm that TWSG1 Ile40 is crucial for BMP ligand binding and demonstrate the significant role of calcium in TWSG1 and BMP ligand interactions. The authors further validate their findings in structure biology by testing them in three distinct elegant systems: cell culture-based reporter assay, organoid assays, and *Drosophila* embryo. Overall, these observations provide outstanding information about TWSG1 structure and its biological significance by a combination of series of experiments. Therefore, this paper has potentially high impact in the fields of structural biology, BMP signaling, developmental biology and medical science. However, several key data are missing to support the novelty of this paper, which need to be addressed prior to publication.

Major points

- The authors have carried out SPR assays in multiple conditions. In Fig. 1F, G the data show that TWSG1 NTD interacts with BMP7 and GDF5 with nanomolar binding constants. I understand that this condition is calcium-free but without EDTA. On the other hand, the authors compare such interactions in conditions of either 2 mM CaCl₂ or 2 mM EDTA in Fig. 3, then conclude that calcium is significant for TWSG1 and BMP ligand binding. Comparing the K_d values between Fig. 1 and Fig.3, K_d in Fig. 1 (calcium-free) is better than the ones in Fig. 3 (2mM CaCl₂). I agree that EDTA negatively impacts TWSG1 and BMP ligand interactions, however, I wonder if 2 mM CaCl₂ may not be an optimal condition for TWSG1 and BMP ligand interactions. I suggest that the authors should test different concentrations of calcium for SPR assays to conclude the significance of calcium for TWSG1 and BMP ligand interactions.
- In Fig. 1H the authors nicely demonstrate that TWSG1 CTD and CHRDL2 interact. Although the authors briefly argue about the ternary TWSG1-BMP-Chordin complex, the biological significance of such interactions can be further confirmed. It would be great if the authors investigate TWSG1 variants (TWSG1, TWSG1 CTD and TWSG1 Ile40Ala) and CHRDL2 complex for binding with BMP ligands using SPR system. It should be very informative when K_d value of TWSG1-CHRDL2 and TWSG1 Ile40Ala-CHRDL2 are tested with BMP ligands. These approaches will unveil how the ternary TWSG1-BMP-Chordin complex affects BMP signaling.
- Given that TWSG1 has not interaction with BMP2 in cell-based assay (Fig. 4F) and not with BMP4 (ref no.30), it has not been described how *Drosophila* Tsg interacts with *Drosophila* BMP ligands. I understand that *Drosophila* Dpp is a BMP4 type ligand. If such interaction paradigm is conserved between vertebrates and insects, how does *Drosophila* Tsg affect BMP signaling without interacting with Dpp ligand? In particular, the interpretation of Tsg I40A or Tsg I40E embryo phenotypes has to be explained by using other approaches as well. If the authors consider that the second BMP ligand Scw is a target of *Drosophila* Tsg, supporting data need to be included. I suggest that the authors should try either SPR assay or cell-based reporter assay using *Drosophila* Tsg variants (Tsg, Tsg I40A and TsgI40E) with *Drosophila* ligands (Dpp and Scw). Such biochemical evidence

would be crucial for interpreting the beautiful *Drosophila* genetics data.

Minor points

- How many samples were tested for the SPR assay (Fig. 1F-H, Fig. 3)? The sample numbers should be described in the figure legends.
- How many samples were tested for the cell-based reporter assay (Fig. 4A-F)? How many samples were tested for the organoid assays (Fig. 4H-K)? The sample numbers should be described in the figure legends.

Reviewer #3 (Remarks to the Author):

In this study by the Siebold group, the authors describe the identification of binding domains for BMPs and the BMP antagonist Chordin within TWSG1. The manuscript is written and logical. The experiments conducted support the conclusions of the paper. However, there are a few weaknesses in the manuscript that would need to be addressed in order for me to accept this paper for publication. A moderate weakness of the paper is that the sample size (n) is not reported for many of the figure panels (for example in figure 3). Statistical analyses are also missing in some of the figure legends. Figure 4 needs a more extensive analysis. The experiments on the C2C12 myoblast cell line rely heavily on the use of a BMP reporter system. At the very least, the authors should demonstrate (through immunofluorescence staining and/or Western blot) that TWSG1 is able to inhibit phosphorylation of SMAD1/5/9 by BMP ligands. In addition, they should provide evidence that TWSG1 has a biological effect on C2C12 by examining myoblast versus osteoblast differentiation (osteoblast differentiation should be inhibited based on BMP response). The colonic organoid experiments are also lacking. The organoid forming efficiency (% of single cells or crypts which go on to form organoids) and the percentage of budding organoids should be quantified with corresponding statistical analyses. Scale bars should be shown with the organoid images. I would recommend resubmission with revisions.

Some minor issues in the text:

147, 184-5 Promoters? Please define.

223 pSMAD staining to show this is happening through canonical BMP signaling?

267 change "buddy" to budding.

279-80 This statement should be reworded, based on the data presented in the paper, since no human cell lines were used for experiments.

Response to the Reviewers

“Molecular Mechanism of BMP Signal Control by Twisted Gastrulation”. Malinauskas, Moore, Rudolf, et. al. - Manuscript # NCOMMS-23-19516

We are immensely grateful to all three Reviewers for their positive feedback on our manuscript, their diligent review process, and the multitude of detailed and constructive comments provided. We have diligently addressed all suggestions, incorporating substantial new experimental data and revising the manuscript accordingly. In instances where acquiring further experimental data was unattainable, we made earnest efforts to provide detailed explanations. We believe the manuscript has undergone significant improvements and is now ready for publication in Nature Communications.

Comments to Reviewer #1

*“The manuscript by Malinauskas, Moore and Rudolf et al., describes a new structure of the BMP modulator twisted gastrulation (TWSG1). **This is a wonderful study which sheds new light into the molecular binding mechanism of TWSG1 and how it interacts with BMP ligands.** The authors show that the N-terminal cys rich domain adopts a helical structure which inserts a helix into the dimer interface to out-compete type I receptor binding. The C-terminus was shown to bind another BMP modulator of the chordin family. Collectively, this information starts to explain how TWSG1, chordin and BMP form an inhibitory complex, however, much is still to be determined and will be a focus of future efforts such as a ternary complex or a complex of the CTD with CHL2. The authors identify a key residue (I40) in the NTD that when mutated abolishes binding. A series of in vivo and in vitro cell assays nicely support the findings. That being said, I have divided the critiques into major and minor comments/concerns which should be addressed by the team of investigators. **Overall, I am very positive for the publication of the study depending on how the concerns are addressed.**”*

1. *“There is a disconnect with the use of the truncated vs full-length TWSG1. Binding experiments were performed with a truncated version and the activity assays were performed with the full-length.”*

SPR-based binding experiments were performed using individual domains of TWSG1 because they are monomeric whereas the full-length TWSG1 tends to dimerize as is shown in our SEC-MALS experiments (**Fig. 1B** and **Supplementary Fig. 1I–J**). For a monovalent, 1:1 binding event, affinity measurements with one partner coupled to a Biacore chip typically mirror the results obtained by isothermal titration calorimetry (ITC), stopped flow fluorescence, or other methods (PMID 23711722 and references therein). But this correspondence breaks down when the species in solution has more than one binding site, as is the case for example when the soluble binding partner is an antibody or the full-length dimeric TWSG1. Bivalent or multivalent binding is a cooperative process, and the apparent binding constant does not reflect a simple 1:1 binding mode, but instead results in a complicated combination of affinity and avidity effects where the apparent K_d is dependent on the ligand concentration on the chip (PMID 23711722 and references therein). For these reasons, we used monomeric, truncated TWSG1, since full-length, dimeric TWSG1, is not an ideal analyte for SPR-based experiments.

2. “How does full-length binding compare to the individual domains and also does the NTD of TWSG1 inhibit BMP signaling to the full potential as TWSG1? While the CTD alone does not bind in SPR experiments it may contribute to binding in the full-length version.”

To address the Reviewer’s comment, we performed additional cellular signaling assays to assess the effect of full length TWSG1 and its individual domains on GDF5 signaling. The NTD of TWSG1 inhibited GDF5 signaling as effectively as the full length TWSG1. In contrast, a construct comprising only the CTD of TWSG1 did not affect GDF5 signaling. Similarly, TWSG1 NTD Ile40Ala, a mutant that impairs binding to BMP ligands, had only a minor effect on GDF5 signaling. These experiments were performed in duplicate, and the results are presented in a new **Supplementary Fig. 7:**

- GDF5 (0 nM)
- GDF5 (30 nM)

Supplementary Fig. 7. The N-terminal domain (NTD) of TWSG1 is sufficient for inhibition of GDF5 signaling in cellular assays. GDF5 (30 nM) activates SMAD-dependent signaling in C2C12 myoblasts (first pink column). Both full-length TWSG1 (1 μM) and its N-terminal domain (NTD, 1 μM) inhibit GDF5 signaling (second and third pink columns, respectively). The mutation Ile40Ala impairs the inhibitory function of TWSG1 Ile40Ala NTD (fourth pink column), whereas TWSG1 CTD has no significant effect on GDF5 signaling (fifth pink column). Each column represents the average GDF5 signaling, measured eight times (n=8, indicated by open circles). Experiments were performed in duplicate in two 96-well plates. Standard deviations are indicated by vertical T-shaped bars on each column. P values were calculated using Student’s two-sample t-test, assuming unequal variance: n.s., not significant, $P > 0.05$; *, $P \leq 0.05$; **, $P \leq 0.01$; ***, $P \leq 0.001$.

3. “SPR traces need to be added for all equilibrium experiments, especially for BMP2. Some were provided in sup figure 2 and some have odd behavior such as panel F.”

We performed all SPR-based binding experiments in duplicate and have now included the sensorgrams for each experiment. The SPR binding isotherms and sensorgrams are shown in the following figures:

1. **Supplementary Fig. 2.** BMP7, GDF5, and BMP2 binding to TWSG1 NTD and CTD.
2. **Supplementary Fig. 4.** BMP7, GDF5, and BMP2 binding to TWSG1 NTD Ile40Ala and TWSG1 NTD Asp34Ala.

4. *“There are significant concerns over the SPR experiments that would take considerable effort to resolve. One limitation of the SPR is that primary amine coupling could be affecting some of the ligands differently. This is certainly the case where the different ligands have a different number of lysine residues. Ideally, one would flip the orientation of the experiment and test ligand binding to a tagged version of the different TWSG1. ”*

We agree with Reviewer #1 that *“that primary amine coupling could be affecting some of the ligands differently”*. We have added the following note in the Methods section to make sure the reader is aware of experimental limitations:

“We acknowledge that coupling BMPs to the SPR chip via primary amines might influence their interactions with analytes, potentially leading to K_d s derived from our experiments differing from those observed in vivo.”

We also note that experiments to measure dissociation constants using dimeric BMPs as analytes and TWSG1 as a ligand immobilized on the chip are not feasible due to the dimerization of BMPs and resulting avidity effects as discussed in our reply to comment 1 by the Reviewer #1, and in PMID 23711722. Thus, we believe that carrying out SPR experiments with BMP ligands as analytes would cause severe problems.

We also note that both BMP2 and GDF5 are not soluble in low-ionic strength buffers at pH levels around 6–7 and concentration (~0.01–100 μ M) necessary for SPR experiments. Typically, BMPs are purified and stored in buffers unsuitable for SPR experiments. For instance, publications from multiple structural biology groups (Thomas Thompson, PMID 27524626; Thomas Mueller, PMID 8620887; Christian Siebold, 32576689, 25938661; Marko Hyvönen, PMID 27036124; Senyon Choe, PMID 20567515; among others) describe storing purified BMP2 and GDF5 at pH levels around 2–4. These research groups rely on similar SPR-based binding experiments with BMPs immobilized via primary lysines on the surface. Considering the limitations of the available experimental approaches to study BMP interactions with their binding partners, we exercise caution in interpreting our SPR-based results. Primarily, we use these results to validate the observed TWSG1–GDF5 interaction interfaces in the crystal structure. For these reasons, we believe that our SPR analysis is the best choice for this study.

5. *“Also, it appears the SPR experiments were only performed a single time with a single (very high) coupling density. A more appropriate experiment would have been to couple a CM5 chip at different densities and perform the equilibrium binding experiment at least twice. In the interest of time and getting this to the public I think the authors need to, at the minimum, provide a replicate for the binding data and include the full length TWSG1 for comparison.”*

We have now performed all SPR-based binding experiments in duplicate and show sensorgrams for all of them as detailed in our response to comment 3 from the Reviewer #1.

6. *“A fair bit of the story includes a Ca binding site for GDF5 (and other ligands) and how Ca enhances binding affinity of TSWG1 to the ligand. Receptor-ligand interactions were tested but not much of an impact was noted. While this might be important, experiments beyond the binding experiments were not included to support the claim that Ca plays a role in the function of TWSG1 or BMP ligands. First, can the authors comment on if a Ca appears in the high-*

resolution structures of other ligands or complexes currently published (perhaps modeled as a water but has similar coordination). “

We have checked all high-resolution (2.5 Å or better) structures of protein–protein complexes containing GDF5 or evolutionarily related BMPs that have the calcium-binding motif identical to GDF5 Gly413-XX-Asp416 (where X is any amino acid residue). These BMPs/GDFs included 11 proteins (BMP2, BMPs 4–7, BMPs 8A/B, BMPs 12 and 13, GDF5, and GDF15) and 12 structures:

1. BMP2–RGMC, PDB ID 4UI1; BMP2–BMPR1A, 1REW, 2QJ9, 2QJB, and 2H62; BMPR1A–BMP2–ActR2, 2GOO;
2. BMP7–Noggin, 1M4U;
3. GDF5–BMPR1B, 3EVS; GDF5–BMPR1A, 3QB4; GDF5–RGMB, 6Z3J; GDF5–RGMC, 6Z3L;
4. GDF15–GFRAL, 5VZ4.

Unfortunately, none of these 12 complexes were crystallized in the presence of calcium, as specified in the “Experiment” section of the respective PDB entries. However, we manually examined the $2F_o-F_c$ and F_o-F_c electron density maps of all 12 complexes. We did not observe any water molecules exhibiting coordination similar to other molecules that might resemble calcium-water interactions observed in the GDF5–TWSG1 crystal structure. We believe additional experiments that are outside the scope of this study are required to investigate the role of the observed calcium binding site.

7. “Secondly, since it appears that Ca binding of the NTD of TWSG1 hinges on aspartate 34, the authors should examine binding of the NTD with D34 to an alanine. Alternatively, one could mutate D416 of GDF5 and test binding. “

We successfully cloned, expressed and purified the TWSG1 NTD Asp34Ala variant as suggested. In duplicate, we performed SPR-based binding experiments to explore the interactions between three BMPs (BMP2, BMP7, and GDF5) and TWSG1 NTD Asp34Ala. Our findings indicate that TWSG1 NTD Asp34Ala binds to the three BMP ligands with K_{d} s ranging from 0.1 to 1 μ M. The SPR sensorgrams and isotherms from these experiments are presented in the new **Supplementary Fig. 4G–L**.

These affinities resemble those observed in BMP–wild type TWSG1 NTD interactions, suggesting that calcium likely plays a minor role in BMP2/BMP7/GDF5–TWSG1 interactions. This aligns with our major discovery, highlighting that Ile40, rather than Asp34, primarily mediates BMP2/BMP7/GDF5–TWSG1 interactions. Consequently, in light of these new results, we have tempered our earlier speculations regarding the potential importance of calcium in mediating interactions between BMPs and their binding partners, and we de-emphasized the role of calcium at several places in the manuscript:

1. We deleted the following sentence from the Introduction: *“This interaction is mediated by calcium, both within the crystal structure and also in solution.”*

2. We have rewritten a paragraph in the Results section on the calcium role in the TWSG1–BMP interactions to reflect the new data: *“Next, we investigated whether calcium contributes to TWSG1–BMP interactions in solution as suggested by our crystal structure (Fig. 2C). We mutated the calcium-binding Asp34 of TWSG1 to alanine and tested binding to BMP7, GDF5 and BMP2 (Supplementary Fig. 4G–L). The TWSG1 mutation Asp34Ala did not abolish BMP interactions,*

resulting in K_{ds} comparable to wild type TWSG1 (**Fig. 1F–H** and **Supplementary Fig. 4G–L**), suggesting that calcium plays a minor role in TWSG1–BMP interactions.”

3. We have de-emphasized the role of calcium for BMP signaling in the Discussion: “*The mutation of the calcium-binding residue Asp34 to Ala in TWSG1 did not abolish the TWSG1–BMP interactions, indicating a minor regulatory role for calcium. Similarly, the interactions between GDF5 and BMPR1B were weakened (though not completely abolished) in the presence of the calcium chelator EDTA*”.

8. “*As a second alternative, the authors might exclude the Ca part of the story which might be better for a follow-up study where there is more room to developed and test the function of Ca in BMP ligands and binding to modulators.*”

Given that mutation of TWSG1 calcium-binding residue Asp34 to Ala had a negligible effect on BMP2/BMP7/GDF5–TWSG1 interactions, we revised the manuscript accordingly. Please see our reply to comment 7 by Reviewer #1.

In addition, we removed SPR-based binding data for BMP2/BMP7/GDF5–BMPR1A ectodomain interactions either in the presence of calcium or EDTA because we have no replicates for these experiments, and BMPR1A is a lower affinity receptor of GDF5 (PMID 15890363).

However, we have performed duplicate SPR-based binding experiments for BMP2/BMP7/GDF5–BMPR1B ectodomain interactions in the presence of calcium or EDTA. BMPR1B serves as the high-affinity receptor for GDF5 (PMID 15890363). Both SPR sensorgrams and isotherms are presented in a new **Supplementary Fig. 5**. We concur with Reviewer #1 that further comprehensive studies are warranted to elucidate the role of calcium in modulating BMP signaling. Nevertheless, we believe that presenting our SPR data for BMP2/BMP7/GDF5–BMPR1B ectodomain interactions is highly relevant to the BMP research community and could serve as a reference for future investigations. Furthermore, our BMPR1B ectodomain construct and its production in mammalian cells establishes a valuable resource for exploring BMPR1B interactions with BMPs in the presence or absence of calcium. To address the Reviewer’s concerns we also included the following sentence in the Results: “*However further studies are required to unravel the complex interactions of BMP ligands with their receptors and modulators*”.

9. “*The specificity of TWSG1 for GDF5 and BMP7 and not BMP2 is interesting. The authors claim that TWSG1 does not inhibit BMP2 (and presumably BMP4), however, a BMP-like inhibition is required in the intestinal organoid assay. The authors use BMP inhibition generically to describe this assay. What BMP is present that needs to be inhibited for the organoid to form? Isn’t it BMP4? If so, can the authors reconcile the specificity of TWSG1.*”

TWSG1 is known to be a poor inhibitor of BMP4 signaling and a good inhibitor of BMP7 signaling in cellular assays, e.g. Fig. 5A-B by Troilo et al., 2016 (PMID 26829466), and our signaling results (TWSG1 inhibits GDF5, BMP7, but not BMP2 signaling) are consistent with these previous observations.

We have performed qPCR assays to identify which BMPs are expressed in mouse intestinal organoids. We discovered that BMP7 is expressed at higher level compared to BMP2 and BMP4. We have included our new qPCR data in the revised manuscript (**Fig. 3H**). Given that TWSG1

inhibits BMP7 signaling in our cell-based reporter assays (and experiments by others, PMID 24586548), it is likely that inhibition of BMP7 signaling is required for the organoids to form. However, we cannot exclude the possibility that signaling by BMP4 and BMP2 plays a role in organoid formation as well.

10. “Also, can the authors suggest a structural explanation for why TWSG1 binds more favorable to GDF5 and BMP7 than BMP2?”

In the manuscript, we prefer not to speculate on why TWSG1 binds more tightly to BMP7 compared to BMP2, considering the limitations of the SPR-based binding experiments as detailed above.

Superposition of apo-BMP7 (PDB ID 1LXI), apo-GDF5 (PDB ID 1WAQ) and the TWSG1–GDF5 complex structures did not reveal any immediate structural explanation. Amino acid sequence identity between the C-terminal signaling domains of BMP7 and GDF5 is 51%, thus multiple residues could be responsible for different binding affinities (**Revision Fig. 1**). We believe detailed studies of the TWSG1–BMP specificity would require years of extensive experimentation and are outside the scope of this manuscript.

Revision Fig. 1. Amino acid sequence alignment of C-terminal signaling domains from human BMP7 (residues Cys330–His431) and GDF5 (Cys400–Arg501).

10. “The exciting experiment where the authors compare the IC50 of BMP signaling with CHL2 alone, TWSG1 alone, CHL2+TWSG1, CHL2+TWSG1 (NTD), CHL2+TWSG1 (CTD) is missing. Since the authors have all the components and cell-based assays this would be straightforward and tie into the data where the authors shown the CTD binds CHL2.”

We performed GDF5 cellular signaling assays using individual domains of TWSG1 and CHRDL2. The results are presented in **Revision Fig. 2** below. CHRDL2, TWSG1 CTD, or TWSG1 CTD+CHRDL2 exhibited minimal effect on GDF5 signaling. TWSG1 efficiently inhibited GDF5 signaling in the presence of CHRDL2. These new findings are consistent with other data presented in the manuscript and emphasize the importance of the NTD (but not CTD) in inhibiting BMP signaling. We believe that more extensive structural studies of Chordin family members and their role in the TWSG1–BMP signaling system are required to shed light on this exciting system; however, we believe that these are outside the scope of this manuscript. Therefore, we would prefer to not include **Revision Fig. 2** in the main manuscript. However, considering the clarity and reproducibility of the data, we are open to including it in the supplementary information, pending guidance from the reviewers and the editor.

- GDF5 (0 nM)
- GDF5 (30 nM)

Revision Fig. 2. TWSG1 efficiently inhibits GDF5 signaling both in the presence and absence of CHRDL2. GDF5 (30 nM) activates SMAD-dependent signaling in C2C12 myoblasts (first pink column). Full-length TWSG1 (1 μM) inhibits GDF5 signaling (second pink column). CHRDL2 (1 μM) had either a minor effect (replicate 1) or no significant effect (replicate 2) on GDF5 signaling (third pink column). TWSG1 efficiently inhibits GDF5 signaling in the presence of CHRDL2 (fourth pink column). TWSG1 CTD fails to inhibit GDF5 signaling either by itself or in the presence of CHRDL2 (fifth and sixth pink columns, respectively). Each column represents the average GDF5 signaling, measured eight times ($n=8$, indicated by open circles). Experiments were performed in duplicate in two 96-well plates. Standard deviations are indicated by vertical T-shaped bars on each column. P values were calculated using Student's two-sample t-test, assuming unequal variance: n.s., not significant, $P > 0.05$; *, $P \leq 0.05$; **, $P \leq 0.01$; ***, $P \leq 0.001$.

11. "A gel showing the quality of all in-house proteins produced and purified should be included (both reduced and non-reduced)."

We analyzed all proteins produced in-house using SDS-PAGE under both non-reducing and reducing conditions. The results of these analyses are displayed in **Supplementary Fig. 12** and below.

Supplementary Fig. 12. SDS-PAGE analysis of proteins used in this study. The samples were heated for 5 minutes at 100 °C before loading onto the gel. Each well contained 1 µg of purified protein. The proteins on the right-side gel were reduced with 2% v/v 2-mercaptoethanol. MW marker, 5 µl/well of BenchMark Protein Ladder from Thermo Fisher Scientific. SDS-PAGE was performed using NuPAGE Bis-Tris 4–12% gel in SDS-MES running buffer (Invitrogen) and stained with InstantBlue Coomassie protein stain (Expedeon). All proteins, except GDF5, BMP2 and Gremlin 1, were produced as secreted proteins via transient expression of HEK293T cells and include full-length glycans.

12. *“(Intro) While sclerostin was originally described as a BMP antagonist, it only inhibits Wnt signaling and this is the mode of action for the Amgen drug, Romosozumab.”*

We have removed the following sentence from the Introduction:

“An anti-Sclerostin antibody, Romosozumab (Amgen), is an FDA-approved drug to treat osteoporosis, highlighting the role of BMP antagonists as drug targets.”

13. *“(Figure 2) Finger 1 and Finger 2 of the ligand are mislabeled.”*

We have corrected labels in **Fig. 2**.

14. *“Why was EDTA and not EGTA used for the Ca chelating experiments?”*

We understand that compared to EDTA, EGTA is more selective for calcium ions over magnesium ions. However, we felt that a more selective EGTA was not essential in this case because no magnesium ion was bound to the TWSG1–GDF5 complex or present in buffers used to purify proteins for SPR-based binding studies.

Comments to Reviewer #2:

“This paper describes the crystal structure of Twisted gastrulation (TWSG1), a conserved BMP binding protein. The authors also investigate the crystal structure of a complex of TWSG1 and GDF5. These studies reveal that the N-terminal domain of TWSG1 forms a complex with BMP ligands, while the C-terminal domain mediates binding with Chordin-like proteins. This study further identifies Ile40 as a key amino acid residue for TWSG1 and BMP ligand interactions. The authors then investigate the protein interactions between TWSG1 and BMP ligands using surface plasmon resonance (SPR)-based equilibrium binding experiments. These approaches confirm that TWSG1 Ile40 is crucial for BMP ligand binding and demonstrate the significant role of calcium in TWSG1 and BMP ligand interactions. The authors further validate their findings in structure biology by testing them in three distinct elegant systems: cell culture-based reporter assay, organoid assays, and Drosophila embryo. Overall, these observations provide outstanding information about TWSG1 structure and its biological significance by a combination of series of experiments. Therefore, this paper has potentially high impact in the fields of structural biology, BMP signaling, developmental biology and medical science. However, several key data are missing to support the novelty of this paper, which need to be addressed prior to publication.”

1. *“The authors have carried out SPR assays in multiple conditions. In Fig. 1F, G the data show that TWSG1 NTD interacts with BMP7 and GDF5 with nanomolar binding constants. I understand that this condition is calcium-free but without EDTA. On the other hand, the authors compare such interactions in conditions of either 2 mM CaCl₂ or 2 mM EDTA in Fig. 3, then conclude that calcium is significant for TWSG1 and BMP ligand binding. Comparing the K_d values between Fig. 1 and Fig.3, K_d in Fig. 1 (calcium-free) is better than the ones in Fig. 3 (2mM CaCl₂). I agree that EDTA negatively impacts TWSG1 and BMP ligand interactions, however, I wonder if 2 mM CaCl₂ may not be an optimal condition for TWSG1 and BMP ligand interactions. I suggest that the authors should test different concentrations of calcium for SPR assays to conclude the significance of calcium for TWSG1 and BMP ligand interactions.”*

All experiments, including those depicted in **Fig. 1**, were conducted in the presence of 2 mM CaCl₂ (or 2 mM EDTA at pH 8.0 where applicable). We apologize for the oversight, which has now been rectified.

We decided to use 2 mM CaCl₂ in our SPR-based experiments because of two main reasons: Firstly, we observed calcium in the TWSG1–GDF5 crystal structure, and we didn’t want to miss out on this potentially important co-factor mediating protein–protein interactions. Secondly, calcium is present in the extracellular space at ~1.0–2.0 mM concentration *in vivo*. We aimed to emulate these conditions as closely as possible in our SPR-based experiments. In the manuscript, we clarified our rationale by changing the statement:

“As the local concentration of calcium in the extracellular space can vary significantly (~1.0–2.0 mM)^{21, 22}, calcium might modulate interactions between BMP family members, their receptors, agonists and antagonists.”

to

“As the local concentration of calcium in the extracellular space can vary significantly (~1.0–2.0 mM)^{21, 22}, calcium might modulate interactions between BMP family members, their receptors,

agonists and antagonists. For this reason, we carried out our SPR-based binding studies in the presence of 2 mM CaCl₂ to reflect extracellular calcium levels”.

However, while revising this manuscript, we conducted further investigations into calcium's role in mediating TWSG1–GDF5 interactions. Our findings revealed that mutating the calcium-binding residue Asp34 to Ala in TWSG1 only had a minor effect on the TWSG1–GDF5 interaction. Consequently, we decided to temper our interpretation of calcium's significance in TWSG1–GDF5 interactions. We have elaborated on these updates regarding calcium in response to Reviewer #1's comments 6 to 8.

2. “In Fig. 1H the authors nicely demonstrate that TWSG1 CTD and CHRDL2 interact. Although the authors briefly argue about the ternary TWSG1-BMP-Chordin complex, the biological significance of such interactions can be further confirmed. It would be great if the authors investigate TWSG1 variants (TWSG1, TWSG1 CTD and TWSG1 Ile40Ala) and CHRDL2 complex for binding with BMP ligands using SPR system. It should be very informative when K_d value of TWSG1-CHRDL2 and TWSG1 Ile40Ala-CHRDL2 are tested with BMP ligands. These approaches will unveil how the ternary TWSG1-BMP-Chordin complex affects BMP signaling.”

Unfortunately, ligands that tend to dimerize (such as full-length TWSG1, CHRDL2, and BMPs) cannot be used as analytes in the SPR-based binding experiments. We have elaborated on these limitations of SPR in response to Reviewer #1's comments 1 and 4.

We furthermore performed GDF5 cellular signaling assays using individual domains of TWSG1 and CHRDL2. Further details regarding these new experiments have been provided in response to Reviewer #1's comment 10.

While we share Reviewer #2's interest and enthusiasm regarding future studies of TWSG1–BMP–Chordin complexes, we believe that comprehensive investigations of these ternary complexes would necessitate years of extensive experimentation and are beyond the scope of this manuscript.

3. “Given that TWSG1 has not interaction with BMP2 in cell-based assay (Fig. 4F) and not with BMP4 (ref no.30), it has not been described how Drosophila Tsg interacts with Drosophila BMP ligands. I understand that Drosophila Dpp is a BMP4 type ligand. If such interaction paradigm is conserved between vertebrates and insects, how does Drosophila Tsg affect BMP signaling without interacting with Dpp ligand? In particular, the interpretation of Tsg I40A or Tsg I40E embryo phenotypes has to be explained by using other approaches as well. If the authors consider that the second BMP ligand Scw is a target of Drosophila Tsg, supporting data need to be included. I suggest that the authors should try either SPR assay or cell-based reporter assay using Drosophila Tsg variants (Tsg, Tsg I40A and TsgI40E) with Drosophila ligands (Dpp and Scw). Such biochemical evidence would be crucial for interpreting the beautiful Drosophila genetics data.”

The potent signaling molecule in the early *Drosophila* embryo is a Dpp–Scw heterodimer. Previously, Tsg has been shown to interact with the Dpp–Scw heterodimer more strongly than Dpp homodimers (PMID 15797386). To gain biochemical evidence relating to the ligand, SPR assays are not feasible as the Dpp–Scw heterodimer has yet to be successfully purified. Therefore, we have used the alternative suggestion of a cell-based assay to test for interactions between wild type and mutant Tsg proteins with the Dpp–Scw ligand, both in the absence and presence of Sog. Results from this assay show that wild type Tsg protein, but not the Tsg Ile40Glu

mutant, can be immunoprecipitated by Dpp–Scw, consistent with the *in vivo* phenotypes observed. We have now included these new data in (**Fig. 4B**).

4. “How many samples were tested for the SPR assay (Fig. 1F-H, Fig. 3)? The sample numbers should be described in the figure legends.”

We performed all SPR-based binding experiments in duplicate and have included the sensorgrams for each experiment. The SPR binding isotherms and sensorgrams are shown in the following figures:

1. **Supplementary Fig. 2.** BMP7, GDF5, and BMP2 binding to TWSG1 NTD and CTD.
2. **Supplementary Fig. 4.** BMP7, GDF5, and BMP2 binding to TWSG1 NTD Ile40Ala and TWSG1 NTD Asp34Ala.

Furthermore, we have incorporated a range of goodness-of-fit statistics for each isotherm in the revised manuscript. These statistics encompass 95% confidence intervals for K_{dS} (dissociation constants), B_{max} (maximum analyte binding), and R^2 (goodness-of-fit of experimental data points to the isotherm model).

5. “How many samples were tested for the cell-based reporter assay (Fig. 4A-F)? How many samples were tested for the organoid assays (Fig. 4H-K)? The sample numbers should be described in the figure legends.”

The cell-based reporter assays presented in the revised **Fig. 3A–F** were performed in duplicate. These duplicates are now displayed in a new **Supplementary Fig. 6**. We have included sample numbers along with selected statistical parameters (95% confidence intervals, R squared, and p values) in the figure and its corresponding legend.

We have repeated experiments with organoids and counted their numbers in the presence of different growth factors. Results are presented in revised **Fig. 3H–N**.

Comments to Reviewer #3:

“In this study by the Siebold group, the authors describe the identification of binding domains for BMPs and the BMP antagonist Chordin within TWSG1. The manuscript is written and logical. The experiments conducted support the conclusions of the paper. However, there are a few weaknesses in the manuscript that would need to be address in order for me to accept this paper for publication. “

1. “A moderate weakness of the paper is that the sample size (n) is not reported for many of the figure panels (for example in figure 3). Stastical analyses are also missing in some the figure legends. Figure 4 needs a more extensive analysis. The experiments on the C2C12 myoblast cell line rely heavily on the use of a BMP reporter system.”

In the revised manuscript, we show duplicates for all SPR-based binding experiments and for all cell-based reporter assays. The SPR binding isotherms and sensorgrams are shown in the following figures:

1. **Supplementary Fig. 2.** BMP7, GDF5, and BMP2 binding to TWSG1 NTD and CTD.
2. **Supplementary Fig. 4.** BMP7, GDF5, and BMP2 binding to TWSG1 NTD Ile40Ala and TWSG1 NTD Asp34Ala.

Furthermore, we have incorporated a range of goodness-of-fit statistics for each isotherm in the revised manuscript. These statistics encompass 95% confidence intervals for K_d s (dissociation constants), B_{max} (maximum analyte binding), and R^2 (goodness-of-fit of experimental data points to the isotherm model).

The cell-based reporter assays presented in the revised **Fig. 3A–F** were performed in duplicate. Duplicates of cellular reporter assays are shown in in a new **Supplementary Fig. 6**. We have included sample numbers along with selected statistical parameters (95% confidence intervals, R squared, and p values) in the figure and its corresponding legend.

2. “At the very least, the authors should demonstrate (through immunofluorescence staining and/or Western blot) that TWSG1 is able to inhibit phosphorylation of SMAD1/5/9 by BMP ligands.”

To adress the Reviewer’s request, we have now used pSmad immunofluorescence and quantitation to show that, consistent with the luciferase assay data, wild type TWSG1 inhibits pSmad activation by GDF5 and BMP7 ligands, but not BMP2 (**Supplementary Fig. 8**). In contrast, the mutant TWSG1 is unable to inhibit GDF5- or BMP7-stimulated pSmad activation.

3. “In addition, they should provide evidence that TWSG1 has a biological effect on C2C12 by examining myoblast versus osteoblast differentiation (osteoblast differentiation should be inhibited based on BMP response).”

We now include data showing that wild type TWSG1, but not the Ile40Glu mutant, inhibits the ability of BMP7 to promote differentiation of C2C12 cells into osteoblasts (**Supplementary Fig. 9**). As such, myotube differentiation is observed in the presence of BMP7 and wild type TWSG1.

4. *“The colonic organoid experiments are also lacking. The organoid forming efficiency (% of single cells or crypts which go on to form organoids) and the percentage of budding organoids should be quantified with corresponding statistical analyses. Scale bars should be shown with the organoid images. I would recommend resubmission with revisions.”*

We have repeated experiments with organoids and counted their numbers in the presence of different growth factors (R-spondin-1, EGF, and Noggin or TWSG1 variants). Results are presented in the revised **Fig. 3H–N**. We have included scale bars in the revised **Fig. 3I–L, 3N** and **Supplementary Fig. 10** with organoid images.

4. *“147, 184-5 Promoters? Please define.”*

We made a mistake in the legend of **Fig. 1C**. We corrected it by changing the following in the revised manuscript:

“(C) Crystal structure of the human TWSG1 dimer with one promoter colored as rainbow (N-terminus, blue; C-terminus, red) and one in grey. The two views differ by a 180° rotation around a vertical axis.”

to

“(C) Crystal structure of the human TWSG1 dimer with one protomer colored as rainbow (N-terminus, blue; C-terminus, red) and one in grey. The two views differ by a 180° rotation around a vertical axis.”

Just for clarity, we used “protomer” appropriately in describing the TWSG1–GDF5 structure in the main text: *“The TWSG1 NTD binds to the finger region of GDF5 and interacts with both protomers of the GDF5 dimer.”*

5. *“223 pSMAD staining to show this is happening through canonical BMP signaling?”*

Please see our reply to comment 2 by the Reviewer #3.

6. *“267 change “buddy” to budding.”*

We corrected our mistake.

7. *“279-80 This statement should be reworded, based on the data presented in the paper, since no human cell lines were used for experiments.”*

We changed *“The mode of TWSG1–BMP interactions is evolutionarily conserved from flies to humans”* to *“The evolutionarily conserved TWSG1–BMP interactions in Drosophila”*.

Reviewer #1 (Remarks to the Author):

The Authors dedicated significant effort to address the comments from the reviewers. Most of the concerns were addressed as best they could in the time frame for the review process. Figure 2 will be left to the discretion of the authors.

While SDS gels were included of the reagents, the authors should be careful with the quality of their ligands, including BMP2 and GDF5 as degradation and a non-reducible forms that pops up from bacterial production are present.

Reviewer #2 (Remarks to the Author):

The reviewer expresses satisfaction with the revised manuscript by Malinauskas et al., noting significant improvements and the high quality of the additional data provided. Overall, the reviewer is pleased with the revisions and the manuscript's focus, which now effectively supports its conclusions. The concerns of all reviewers have been addressed, and the manuscript is considered greatly improved.

Reviewer #3 (Remarks to the Author):

Although the authors have addressed most of my concerns, I still have a few suggestions and concerns based on their new data. Supplementary figures 8 and 9 could be improved. In Supplementary figure 8, what is the pSMAD antibody used? The figure is just labeled pSMAD and the only pSMAD antibody listed in Supp Table 2 is a pSMAD3 antibody. Although BMP signaling can induce SMAD2/3 phosphorylation (which functions downstream of TGF- β), BMPs mainly signal through pSMAD1/5/9. I would recommend that the authors repeat these experiments and examine pSMAD1/5/9. I would recommend using this antibody from Cell Signaling Technologies- Phospho-SMAD1 (Ser463/465)/ SMAD5 (Ser463/465)/ SMAD9 (Ser465/467) (D5B10) Rabbit mAb #13820. Also, I would recommend presenting the pSMAD1/5/9 channel in red and showing the overlap with DAPI so that it becomes obvious that pSMAD1/5/9 is nuclear. Supplementary figure 9 is also good but there is not quantification of the Myosin heavy chain IV or the Alk-Phos expressing cells. At the very least they should quantify the relative pixels in each channel. Also, the Alk-Phos should be show in red so that it easier to see the positive cells The magenta color used is difficult to see. I would gladly accept the manuscript contingent on these corrections.

Molecular Mechanism of BMP Signal Control by Twisted Gastrulation

Malinauskas, Moore, Rudolf, *et. al.*, 2023

Manuscript # NCOMMS-23-19516A

12 April 2024

Response to the reviewers

We are immensely grateful to all three Reviewers for their positive feedback on our manuscript, their diligent review process, and the multitude of detailed and constructive comments provided. We have addressed final suggestions by Reviewer #3. We believe the manuscript is now ready for publication in Nature Communications.

Reviewer #1

The Authors dedicated significant effort to address the comments from the reviewers. Most of the concerns were addressed as best they could in the time frame for the review process. Figure 2 will be left to the discretion of the authors.

While SDS gels were included of the reagents, the authors should be careful with the quality of their ligands, including BMP2 and GDF5 as degradation and a non-reducible forms that pops up from bacterial production are present.

We thank Reviewer #1 for her/his support and detailed, constructive comments.

We have double-checked Figure 2 and believe it is suitable for publication.

Similarly, we believe that the quality of our ligands (BMP2 and GDF5) is suitable for structural and functional studies, as suggested by their biological activity in a panel of cellular assays and their ligand-binding properties in SPR-based binding experiments. Structural integrity of GDF5 has been confirmed by X-ray crystallography. The degradation level of BMP2 and GDF5 observed under reducing conditions on SDS-PAGE is negligible (Supplementary Fig. 12).

Reviewer #2

The reviewer expresses satisfaction with the revised manuscript by Malinauskas et al., noting significant improvements and the high quality of the additional data provided. Overall, the reviewer is pleased with the revisions and the manuscript's focus, which now effectively supports its conclusions. The concerns of all reviewers have been addressed, and the manuscript is considered greatly improved.

We thank Reviewer #2 for her/his enthusiastic support and detailed, constructive comments.

Reviewer #3

Although the authors have addressed most of my concerns, I still have a few suggestions and concerns based on their new data. Supplementary figures 8 and 9 could be improved. In Supplementary figure 8, what is the pSMAD antibody used? The figure is just labeled pSMAD and the only pSMAD antibody listed in Supp Table 2 is a pSMAD3 antibody. Although BMP signaling can induce SMAD2/3 phosphorylation (which functions downstream of TGF- β), BMPs mainly signal through pSMAD1/5/9. I would recommend that the authors repeat these experiments and examine pSMAD1/5/9. I would recommend using this antibody from Cell Signaling Technologies- Phospho-SMAD1 (Ser463/465)/ SMAD5 (Ser463/465)/ SMAD9 (Ser465/467) (D5B10) Rabbit mAb #13820. Also, I would recommend presenting the pSMAD1/5/9 channel in red and showing the overlap with DAPI so that it becomes obvious that pSMAD1/5/9 is nuclear.

Supplementary figure 9 is also good but there is not quantification of the Myosin heavy chain IV or the Alk-Phos expressing cells. At the very least they should quantify the relative pixels in each channel. Also, the Alk-Phos should be show in red so that it easier to see the positive cells The magenta color used is difficult to see. I would gladly accept the manuscript contingent on these corrections.

We thank Reviewer #3 for her/his support and additional suggestions.

The pSMAD3 antibody we used is the one listed in Supplementary Table 2, which recognises pSMAD1/5/9 as well as pSMAD2/3 (DOI:10.1126/scitranslmed.aaf1090). For this reason, we labelled Supplementary Fig. 8 with the more generic 'pSMAD' label. However, we consider it more likely that we are detecting pSMAD1/5/9 since, as the Reviewer points out, BMPs mostly signal through SMAD1/5/9. Using this antibody allows us to detect all phosphorylated SMAD proteins, which reveals that wild type TWSG1, but not the Ile40Glu mutant, inhibits SMAD phosphorylation downstream of BMP in C2C12 cells.

Please note that we do discuss the pSMAD3 antibody cross-reactivity in the Methods (lines 677/678 in the manuscript version NCOMMS-23-19516A):

"pSmad was visualised by staining with a rabbit anti-Smad3 (phospho S423+425) PUR (1:750 dilution) primary antibody that cross-reacts with phospho-Smad1⁶⁹ and a donkey anti-rabbit IgG Alexa Fluor 647 (1:500) secondary antibody."

Reference 69. Dey D, et al. Two tissue-resident progenitor lineages drive distinct phenotypes of heterotopic ossification. Sci Transl Med 8, 366ra163 (2016).

We do not consider that the other changes Reviewer #3 requested relating to the colours or quantitation are essential for interpretation of the data presented. In fact, we purposely avoided the red/green colour combination that the Reviewer suggests in their final point, as this would make the figure inaccessible to colour blind readers.